# Understanding Graph Self-Supervised Pre-training under Distribution Shifts: A Scaling Law Perspective

## Abstract

Scaling laws have played a fundamental role in the development of foundation models for NLP and vision, but their applicability to large-scale pretrained graph-based models remains unclear—particularly under distribution shifts intrinsic to graph data. In this work, we systematically investigate how model capacity and data scale affect downstream performance in graph pre-training under distribution shifts. To disentangle how distribution shifts impact the scaling, we construct synthetic benchmarks based on contextual stochastic block models, with precise control over both structural and feature-level shifts across the pre-training and testing graphs. Our initial experiments on GCN, a standard Graph Neural Network (GNN) baseline, reveal a striking asymmetry: increasing model capacity consistently improves performance, while increasing data size often degrades it, even under mild shift. We show that this degradation is not inevitable; properly configuring the pretraining model with deeper, wider, and transformer-based architectures enables favorable data scaling, even when distribution shifts. As data scales, graph transformer models achieve up to +9% gains over GCN, which holds for both synthetic and real-world graph domain adaptation tasks. To explain this phenomenon, we develop a theoretical framework based on Fisher separability and Wasserstein domain divergence, which formally characterizes how distribution shifts affect representation transferability. Our results highlight architecture- and shift-aware strategies as the key to unlock scalable graph-based model pre-training.

## 1 Introduction

Graph neural networks (GNNs) have become a cornerstone for learning on graph-structured data, achieving success in tasks such as node classification (Kipf, 2016; Veličković et al., 2017; Rong et al., 2019; Luo et al., 2024), link prediction (Zhang & Chen, 2018; Rossi et al., 2021; Liu et al., 2025b), and molecular property prediction (Wang et al., 2022; Xia et al., 2023). Recently, self-supervised pre-training has emerged as a promising strategy for scaling GNNs to diverse real-world problems (Hu et al., 2019; Qiu et al., 2020; Jiang et al., 2021; Lu et al., 2021; Xu et al., 2023; Liu et al., 2023b; Yu et al., 2024a; Zhao et al., 2024a; Yu et al., 2025b; Liang et al., 2025). Inspired by the success of pre-training in natural language processing and computer vision, this line of work aims to learn transferable graph representations from large collections of unlabeled data.

With the advent of large-scale pre-training, a central question arises: does increasing model capacity and training data reliably improve downstream performance in graph learning? In NLP and CV, this question has been systematically addressed through the study of scaling laws (Kaplan et al., 2020; Zhai et al., 2022), which show that model performance improves in predictable ways as models and datasets grow. These empirical rules offer practical heuristics for building large models coupled with theoretical insight into the dynamics of deep learning at scale (Bi et al., 2024; Liu et al., 2025a). The existence of scaling laws which are analogous for GNNs is not yet established.

Indeed, recent studies have challenged the universality of scaling laws in the graph domain. For instance, Xu et al. (2023) pre-trained GNNs on 11 real-world datasets and found that performance does not exhibit a consistent trend with increasing data size, even degrading in some instances. Similarly, Ma et al. (2024); Wang et al. (2024b); Pengmei et al. (2024) reported the absence of clear

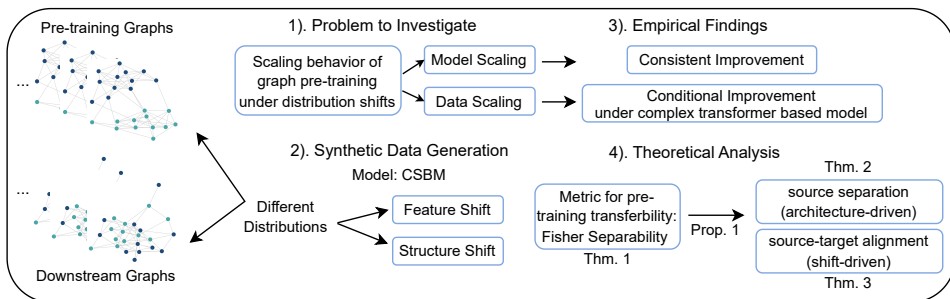

Figure 1: Overview of our study on graph pre-training scaling behavior under distribution shift. We begin by identifying the central question—how model size and data size affect downstream performance when the source and target graphs follow different distributions. To isolate these factors, we construct a synthetic benchmark based on CSBM that enables controlled feature-level and structure-level shifts. This benchmark allows us to isolate the empirical scaling phenomena—most notably the striking asymmetry between model scaling and data scaling—and motivates the development of a theoretical framework that explains how structure–feature alignment and architectural properties jointly shape the transferability of Graph Pre-training.

scaling behavior in large-scale graph classification and node classification tasks. These findings cast doubt on the assumption that larger data or model size inevitably leads to improved performance in GNNs. As such, graph representation learning lacks clear guidance for model design and solid theoretical foundation for scalability.

In this work, we aim to uncover scaling behavior in graph pre-training. To this end, establishing such laws for graphs is substantially more challenging than with text or images, this is due to two intertwined obstacles that confound the analysis. First, distribution shift is an intrinsic challenge. Real-world graphs differ widely in structural topology and feature distributions, hindering the transferability of pre-trained representations and obscuring the emergence of consistent scaling trends (Huang et al., 2024; Wang et al., 2024a). Second, data scarcity remains a critical barrier. The graph domain lacks the expansive, diverse, and high-quality datasets that have enabled the identification of scaling laws in other areas (Bechler-Speicher et al., 2025), raising the question of whether the absence of scaling is due to intrinsic limitations of graph models or the insufficiency of available data.

Given this lack of clarity, we construct synthetic benchmarks based on the contextual stochastic block model (CSBM) (Deshpande et al., 2018; Baranwal et al., 2022; Zheng et al., 2024), designed with varied distribution shifts and data scales to systematically analyze the effect of model capacity and dataset size on GNN-scaling behavior. Specifically, we apply CSBM as a means for precise control over both structural and feature-level shifts within graph data. Our benchmarks include one clean baseline (Shift 0) and five variants (Shifts 1–5) that span mild and severe feature shifts, as well as structure shifts arsing from introducing graph heterophily and edge perturbations.

Our initial investigations with GCN reveal a striking asymmetry in how GNNs scale between their model and input data size. For model size, **Increasing GNN layer depth or width consistently improves downstream performance** across all datasets, even under substantial distribution shifts. In contrast, **increasing the size of the pre-training data often degrades downstream accuracy**, including in settings with mild shift (e.g., Shift 1). This surprising result suggests that distribution shift alone cannot fully account for the failure of data scaling in GNNs.

To further investigate scaling behavior, we examine the role of model capacity and backbone type. Our analysis shows that **data scaling is highly sensitive to design choices**: shallow or narrow GNNs often degrade with more data, whereas deeper, wider, and Transformer-based architectures exhibit more stable and favorable scaling. This pattern holds consistently across synthetic benchmarks and real-world graph domain adaptation tasks, where graph transformer model achieves up to +9% accuracy gains over message-passing counterparts. These results underscore architecture as a critical yet often overlooked determinant of scalable graph model pre-training (Luo et al., 2024).

To understand these empirical phenomena, we develop a theoretical framework based on Fisher separability (Fisher, 1936) and Wasserstein domain divergence (Villani et al., 2008; Cuturi, 2013). This framework formally characterizes how distribution shifts degrade representation transferability in self-supervised graph pre-training. It also explains when and why increased data size may lead to negative transfer, despite improved model capacity. Our analysis highlights the critical role of graph structure and feature alignment in determining transfer success, offering a principled explanation for the asymmetries observed in our experiments. These insights offer both theoretical justification and practical guidance for designing scalable and robust graph pre-training pipelines. Figure 1 provides a brief overview of our workflow.

## 2 RELATED WORK

**Graph Self-Supervised Learning and Foundation Models.** Self-supervised learning (SSL) has become central in graph representation learning through enabling GNN pre-training without labeled data. Graph-based SSL techniques include, the contrastive approach (Veličković et al., 2018; You et al., 2020) and recent masked modeling like GraphMAE (Hou et al., 2022) which reconstruct masked node features and structures. Despite improved benchmarking performance, little attention has been paid to understand the scaling behavior for graph SSL. Likewise, inspired by large language models, there is growing interest in Graph Foundation Models (GFMs) trained at scale (Lachi et al., 2024; Yu et al., 2024b;c; Zhao et al., 2024b; Wang et al., 2025; Zhao et al., 2025), which show transfer potential, particularly in molecular tasks (Lachi et al., 2024; Méndez-Lucio et al., 2024). However, transfer remains challenging: scaling pre-training data does not always improve accuracy and may even cause negative transfer under heterophily or feature noise (Xu et al., 2023; Wang et al., 2024a; Huang et al., 2024). This raises a key question: under what conditions does self-supervised pre-training help or hurt performance, particularly with respect to model size, data scale, and distribution shift? (More discussions in App. C)

**Neural Scaling Laws in Deep Learning.** Scaling laws—empirical relationships between model performance, data size, and model capacity—have become a central theme in understanding and designing large foundation models. Seminal works (Hestness et al., 2017; Kaplan et al., 2020; Henighan et al., 2020; Zhai et al., 2022) show that in language and vision tasks, performance improves predictably with scale, following a power-law trend. These insights have shaped the development of large models such as GPT-3 (Brown et al., 2020) and PaLM (Chowdhery et al., 2023). In the graph domain, there is no consensus on whether similar trends hold for node classification (Xu et al., 2023; Pengmei et al., 2024; Liu et al., 2024; Ma et al., 2024). Moreover, prior studies have not considered how distribution shift—ubiquitous in real-world graphs—affects GNN scaling dynamics.

**Theoretical Perspectives on Graph Transfer and Domain Adaptation.** Prior theoretical work analyzed GNN generalization under distribution shifts on structural perturbations (Liu et al., 2023a; Fang et al., 2025b), spectral misalignment (Pang et al., 2023; You et al., 2023), and feature shifts (Cai et al., 2024; Fang et al., 2025a). However, these studies largely consider supervised settings with fixed architectures, offering little insight into how performance evolves during self-supervised pre-training.

## 3 MAIN ANALYSIS

### 3.1 EXPERIMENTAL SETUP AND DATASET GENERATION

**Training and Transfer Protocol** We focus on the node classifcation task and adopt the standard two-stage training protocol, as shown in (Yu et al., 2024c; Zhao et al., 2024a; Yu et al., 2025a; Liang et al., 2025). In the pre-training stage, a self-supervised GNN encoder with GCN as backbone is trained on $N_S$ unlabeled graphs. In the linear probing stage, the linear classifier is trained with frozen encoder using $m$-shot labeled nodes per class optimized with cross-entropy loss, and evaluated on the remaining nodes, with a total of $N_T$ target graphs. Our initial investigation applied a variety of representative SSL methods (Hou et al., 2022; Kipf & Welling, 2016; Veličković et al., 2018; You et al., 2020), resulting in similar observations. Given these observations, the following sections use GraphMAE as an example. The remaining examples can be found in App. B.

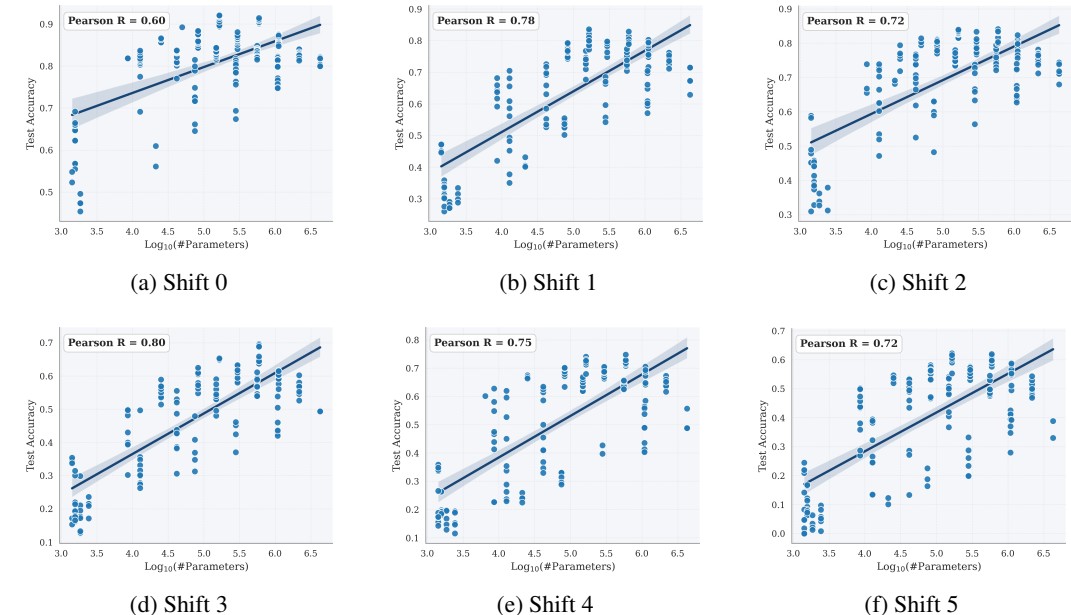

(a) Shift 0          (b) Shift 1          (c) Shift 2

(d) Shift 3          (e) Shift 4          (f) Shift 5

Figure 2: **Model scaling under distribution shifts: Parameter count vs. test accuracy.** Each subplot shows test accuracy versus model size ($\log_{10}$(#parameters)) under a specific type of distribution shift (Shift 0–5). Each point represents a trained model with a specific depth and width. The fitted line indicates the overall scaling trend, and the Pearson correlation $R_{\text{model}}$ summarizes its strength.

**Data Generation** Each graph contains $n$ nodes (default $n = 1000$), evenly assigned to $C$ classes, where each node is associated with a feature vector in $\mathbb{R}^d$. To systematically study the behavior of GNN pre-training under distribution shifts, synthetic graphs are generated through a three-step process: *label assignment*, *feature construction*, and *structure generation*, with varying distribution shifts across the pre-training and downstream testing graphs.

• *Label Assignment and Feature Construction.* We assign class labels across nodes uniformly and construct node features based on their assigned class. For each class $c$, we sample a prototype vector $\mu_c \in \mathbb{R}^d$ uniformly from the unit sphere and scale it to some fixed norm $\sigma$. Each node $i$ is then assigned a soft class distribution $\pi_i \in \mathbb{R}^C$ and its feature is a convex combination of class prototypes: $x_i = \sum_{c=1}^{C} \pi_{i,c} \cdot \mu_c$. We define various shifts to systematically vary the level of feature ambiguity and inter-class separability.

**Shift 0** serves as the baseline, where each node is assigned a fixed soft label[1] vector depending on its class. **Shift 1** increases the difficulty by assigning soft labels that reduces inter-class separation, introducing stronger class ambiguity. **Shift 2** uses the same soft label distributions as Shift 0, but reduces the inter-class separation by scaling $\sigma$ from a larger $\sigma_1$ to a smaller $\sigma_2$. For **Shifts 3–5**, the feature distribution is kept identical to Shift 0, allowing us to isolate and examine structural shifts.

• *Graph Structure.* Graph structures are generated using the contextual stochastic block model (CSBM) (Deshpande et al., 2018), where the probability of forming an edge between two nodes depends only on their class labels. For scenarios emphasizing feature shifts (Shift 0–2), we use a fixed block probability matrix with intra-class connection probability $p_{\text{in}}^{\text{high}}$ and inter-class probability $p_{\text{out}}^{\text{low}}$, yielding homophilic graphs. To simulate structural distribution shifts (independent of feature shift), we introduce the following modifications in **Shift 3-5**. In **Shift 3**, we reduce edge homophily by lowering the intra-class connection probability $p_{\text{in}}^{\text{low}}$ and increasing inter-class connection probability $p_{\text{out}}^{\text{high}}$, resulting in weaker community structure and noisier graphs. In **Shift 4**, we maintain the base block probability matrix, but induce two types of noise: (1) the edge probabilities are randomly perturbed on a per-graph basis; (2) edges are randomly deleted (10%) and new edges are inserted (5%) to simulate edge-level corruption. **Shift 5** combines Shifts 3 and 4, introducing changes in homophily along with edge perturbations, producing the noisiest and most structurally challenging scenario. All parameter choices are listed in App. B.3.

---

[1]Here we provide additional discussion on scaling behavior.

**Scaling Analysis Setup** We analyze how downstream performance scales with two key factors: **model size**, controlled via the number of GNN layers $L \in \{2, 4, 8, 16\}$ and hidden dimensions $d \in \{8, 32, 64, 128, 256\}$ (spanning $10^3$ to $10^7$ parameters), and **data size**, given by the number of pretraining graphs $N \in \{100, 1,000, 10^4, 5 \times 10^4, 10^5, 5 \times 10^5\}$. Each configuration is pretrained and evaluated using the same 25-shot linear probing protocol introduced in Section 3.1. Performance is averaged over 5 random seeds. To quantify scaling trends, we compute Pearson correlations between test accuracy and the log-scaled variables: $R_{\text{model}} = \text{corr}(\text{Acc}, \log_{10}(\#\text{Parameters}))$, $R_{\text{data}} = \text{corr}(\text{Acc}, \log_{10}(N))$. These metrics capture monotonic relationships without assuming a specific functional form.

## 3.2 How Does Model and Data Scaling Behave under Different Distribution Shifts?

To uncover the neural scaling behavior of GNNs under realistic variations, we conduct a fine-grained analysis of how model and data size impact test accuracy across different types of distribution shifts. We pretrain the GCN model using GraphMAE as discussed in Sec. 3.1 as default setting for investigation. Our investigation focuses on six synthetic shifts(including a no-shift baseline (Shift 0) and five shifted scenarios (Shift 1-5)) with increasing deviation in structure or features. We report Pearson correlations between accuracy and scaling variables, supported by visualizations that reveal both global and architecture-sensitive trends.

**Can Model Scaling Help Under Distribution Shift?** Figure 2 plots test accuracy against model size across all shifts. Each dot corresponds to a unique (layers, hidden dim) configuration. We observe a strong and consistent positive correlation between model size and performance across nearly all shifts.

From Fig. 2, under **Shift 0** (no shift), we observe $R_{\text{model}} = 0.60$, indicating only modest gains since small models suffice. For **Shifts 1–2** (feature shifts), $R_{\text{model}} = 0.78$ and $0.72$, showing that larger models better capture misaligned or complex feature–label relations. Under **Shifts 3–5** (structural shifts), $R_{\text{model}} \in [0.72, 0.80]$, confirming that scaling remains beneficial despite heterophily, randomness, and edge noise. These results demonstrate that **GNN pre-training benefits consistently from increasing model capacity**, regardless of the type or severity of distribution shift.

**Can Data Scaling Help Under Distribution Shift?** We next examine how increasing the amount of pre-training data affects downstream performance. Figure 3 plots test accuracy versus the log-scaled number of pre-training graphs across all Shifts. Unlike model size, which shows a clear positive trend, **data scaling yields inconsistent or even negative results**.

- **Shift 0 (No shift):** When pre-training and downstream data are aligned, increasing data scale provides marginal benefit ($R_{\text{data}} = 0.11$). This result suggests that data scaling may suffer from early saturation or introduce redundancy, even without distribution shifts.
- **Shifts 1–2 (With feature shifts):** Under the presence of feature shifts, the correlation turns slightly negative with around $-0.10$, indicating that more graphs with misaligned features might make it more difficult for the model to learn class-consistent representations.
- **Shifts 3–5 (With structure shifts):** Compared to feature shifts, structure shifts incur more negative impacts when scaling up the data ($R_{\text{data}} \in [-0.46, -0.25]$). Specifically, with increasing structure shift from shift 3 to 5, increasing pre-training data only worsens performance by injecting mismatched or noisy samples, reducing representation transferability.

These findings reveal a key asymmetry in GNN scaling: **while model capacity consistently improves performance, increasing data scale can hurt transfer—especially under distribution shift**. However, this does not imply that data scaling is fundamentally harmful. In later sections, we show that the success of scaling pre-trained GNNs depends critically on model configuration and architectural robustness, particularly in the presence of structural shift.

## 3.3 When Does Data Scaling Help? The Role of Model Capacity.

To understand when data scaling is effective, we analyze the Pearson correlation between test accuracy and the logarithm of pre-training data size across different model configurations (Figure 4). For consistency with Section 3.2, we use the same set of models; only varying in depth and width.

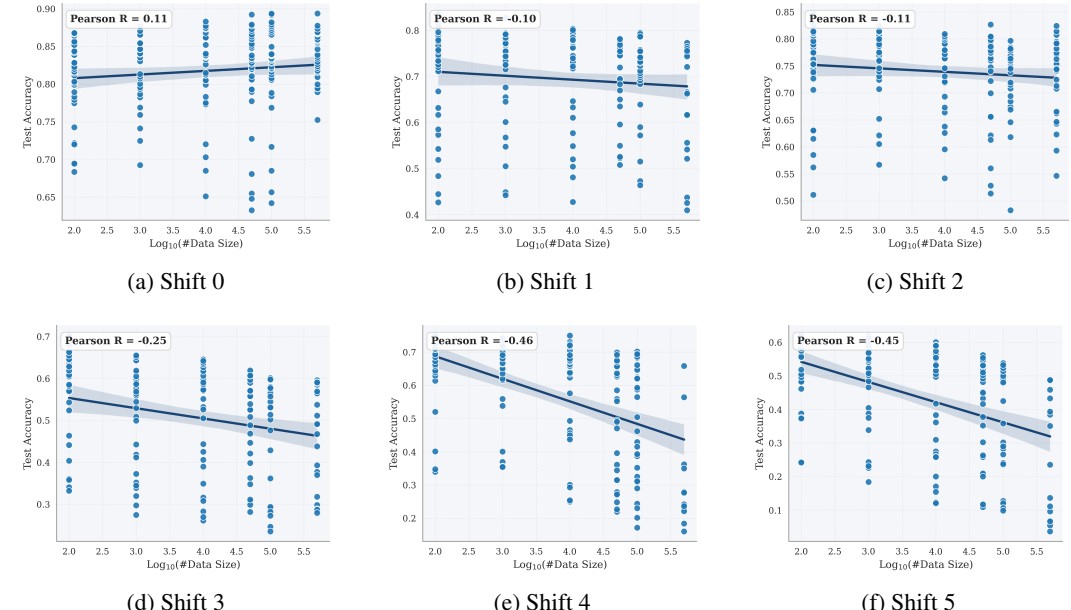

Figure 3: **Data scaling under distribution shifts: Data size vs. test accuracy.** The subplots shows test accuracy versus pre-training data size ($\log_{10}(N)$) under Shift 0–5. Each point represents a trained model with a specific depth and width. The fitted line indicates the overall trend, and the Pearson correlation $R_{\text{data}}$ summarizes its strength.

The key difference is that Figure 3 reports overall trends aggregated across models, while here we disentangle the effects by model capacity. We find that **only sufficiently expressive models—those with both large depth and width—consistently benefit from increasing data**, particularly under distribution shift.

Figure 4 shows that the effect of data scaling depends critically on model size. Under **severe structural shifts** (Shifts 3–5), only deep and wide GNNs achieve positive correlations with data size. For example, in Shift 5, architectures with 16 layers and 256–512 hidden units yield $R_{\text{data}} > 0.5$, while smaller models consistently show negative scaling. This indicates that high capacity is necessary to denoise corrupted edges and recover transferable structure. A similar pattern holds in feature shift with Shifts 1–2, where large models (e.g., 8 layers, 256 hidden units and larger) achieve stronger positive scaling ($R_{\text{data}} > 0.3$), suggesting that complex feature distributions require greater depth and width to capture high-variance signals.

By contrast, in **no-shift settings** (Shifts 0), shallow models already perform well, and additional capacity provides little to no benefit. Even 2–4 layer networks achieve near-maximal accuracy, consistent with oversmoothing effects (Loukas, 2019; Chen et al., 2020).

These findings establish a clear design principle: **effective data scaling under distribution shift requires proportionally increasing model depth and width**. Without sufficient capacity, additional pre-training data can offer negligible or even negative returns. Thus, the benefits of data scaling are conditional—not guaranteed—and tightly coupled to model expressiveness and the nature of distribution shift.

### 3.4 Do GNNs and Transformers Scale Differently Under Distribution Shift?

While the previous sections show that larger models improve transferability, we now ask: *do all architectures benefit equally from scaling under distribution shift?*. To investigate, we compare four representative architectures—GCN, GraphSAGE (Hamilton et al., 2017), GAT (Veličković et al., 2017), and graph transformer (Chen et al., 2022)—under identical pretraining configurations with hidden_dim $d = 512$, GNN layer number $L = 16$ in Figure 5(a, b) and pretraining graphs number $N = 1000$ in Figure 5(b) . For each model, we measure the Pearson correlation between log-scaled data size and downstream accuracy, and also track absolute accuracy gains from scaling.

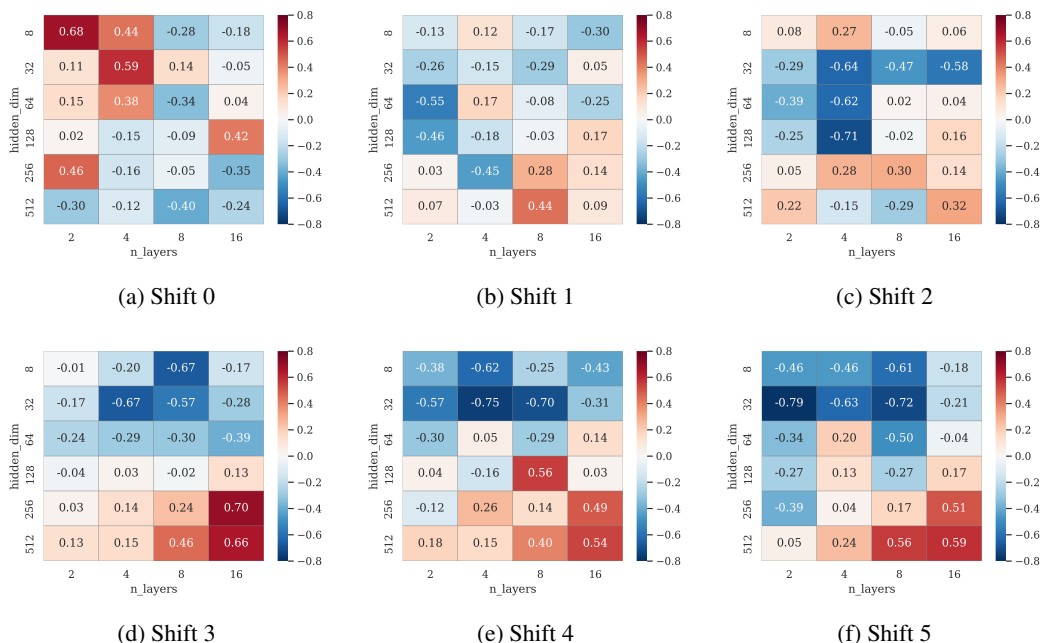

Figure 4: **Impact of Model Size on Data Scaling under Distribution Shifts.** Each heatmap shows Pearson $R$ between test accuracy and log-data size for different architectures (rows: hidden dimension, columns: layers) across six shift scenarios. Warmer colors indicate stronger positive scaling effects.

**Graph Transformer Scales More Effectively Under Shift.** As shown in Figure 5(a), Graph transformer consistently exhibits the **strongest positive correlation** between data size and performance across all Shifts. In Shift 0 (no shift), their Pearson correlation is already 0.20—above all other models. In high-shift scenarios like Shift 5, the correlation increases to 0.84, while GCN and GraphSAGE remain below 0.7. This suggests that graph transformers are better able to absorb information from additional unlabeled data under distribution shifts.

**Scaling Gaps Widen with Distribution Shift.** Figure 5(b) shows that the **performance gap between graph transformer model and other GNNs grows with shift severity**. In Shift 0, the gain of Transformers over GCN is minimal (~2%), but in Shift 5, this margin exceeds 14%. Similar trends are observed for GraphSAGE and GAT. This widening gap highlights the importance of architectural inductive biases—particularly global reasoning—in enabling effective scaling under shift.

While most models benefit from data increases, **only graph transformer model demonstrate stable scaling gains across all shift levels**. Their stronger correlation with data size and growing margins over other GNNs confirm that architecture plays a central role in scalable graph pretraining.

### 3.5 DO THE OBSERVED SCALING TRENDS GENERALIZE TO REAL-WORLD GRAPHS?

To test whether the scaling behaviors from synthetic benchmarks hold in practice, we evaluate on two real-world graph domain adaptation tasks built from the Airport (Ribeiro et al., 2017),: USA → EUROPE and EUROPE → BRAZIL (Wu et al., 2020; Liu et al., 2023a; Fang et al., 2025a) These tasks involve substantial cross-domain distribution shifts. The Airport dataset consists of airport networks from three regions—USA (U), Europe (E), and Brazil (B)—each representing a distinct structural and feature distribution. These settings exhibit substantial cross-domain distribution shifts. The dataset contains airport networks from three regions—USA (U), Europe (E), and Brazil (B)—each with distinct structural patterns and feature statistics. For each task, we fix the target graph for evaluation and pretrain models on varying fractions (20%–100%) of the source graph using GraphMAE (Hou et al., 2022). All architectures—GCN, GraphSAGE, GAT, and graph transformer —are trained with the same objectives and pipelines, without access to source-domain labels. Results are reported in Figure 6.

**Transformer models retain scaling benefits under realistic shifts.** Across both transfer settings, Graph Transformers exhibit the most consistent and monotonic gains as the fraction of pre-training

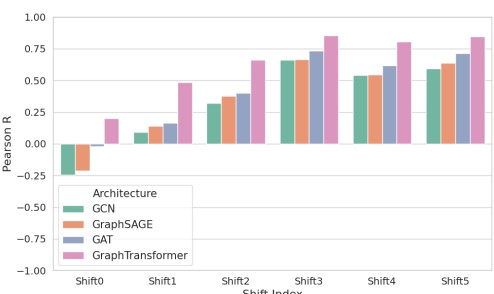 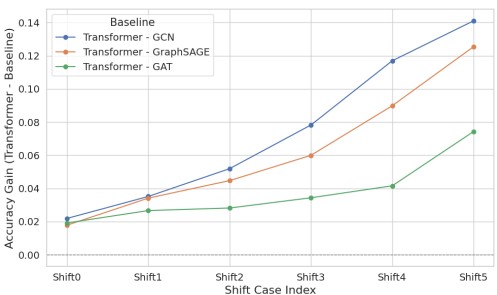

(a) Pearson correlation $R_{\text{data}}$ between $\log_{10}(N)$ and Accuracy

(b) Accuracy Gain of GraphTransformer Over GCN, GraphSAGE, and GAT

Figure 5: **Scaling trends across architectures under distribution shift.** `hidden_dim = 512`, `n_layers = 16`, **(Left)**: Pearson correlation $R_{\text{data}}$ between $\log_{10}($data size$)$ and test accuracy, averaged repeated 5 runs. Transformers show the most consistent and strongest scaling trend. **(Right)**: Accuracy gain of graph transformer over GCN, GraphSAGE, and GAT. Gaps widen with shift ~~~~~~~~ for ~~~~~~

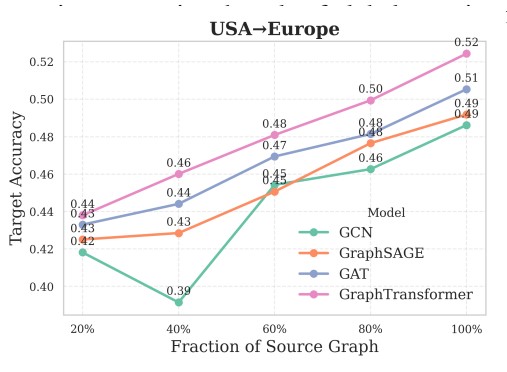 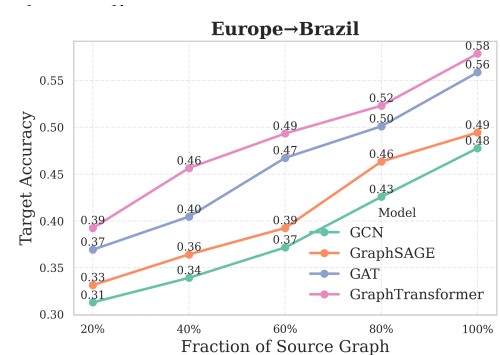

(a) Target accuracy on the USA → EUROPE task across different source graph fractions.

(b) Target accuracy on the EUROPE → BRAZIL task across different source graph fractions.

Figure 6: **Graph transformer consistently scales under real-world domain shifts. (a)** Accuracy trends on the USA → EUROPE task. **(b)** Accuracy trends on the EUROPE → BRAZIL task.

data increases. In EUROPE→BRAZIL, accuracy improves steadily from 0.39 at 20% of the source graph to 0.58 at 100%, while GCN and GraphSAGE rise more slowly and with less stability. A similar trend appears in USA→EUROPE, where Graph Transformers reach 0.52, outperforming GCN and GraphSAGE at 0.49 each. These results mirror our synthetic benchmarks and highlight that graph transformers are uniquely effective at leveraging larger pre-training datasets under distribution shifts.

**Architecture-performance gaps widen under shift.** The performance gap between graph transformer and GCN grows from 2% in USA → EUROPE to 10% in the more challenging EUROPE → BRAZIL setting. This echoes our earlier observations: scaling effectiveness is not solely determined by data quantity, but critically depends on the inductive bias of the architecture in handling distribution shifts.

These results validate that the scaling dynamics observed under controlled synthetic shifts not only persist in real-world settings, but in fact become more pronounced under naturally occurring distribution shifts.

## 4 THEORETICAL ANALYSIS: LIMITS AND DRIVERS OF PRETRAINING TRANSFERABILITY

In this section, we develop a unified theoretical framework that reveals how model capacity, architectural design, and domain shift interact to determine the transferability of GNN pretraining. In particular, we connect these factors through Fisher separability (Fisher, 1936) and Wasserstein distance (Villani et al., 2008; Cuturi, 2013), providing a principled explanation of the scaling laws

observed in previous sections. Due to limited space, we only present the condensed theorems in this section. More details and proofs can be found in App. A.

**Fisher Separability as a Metric for Pretraining Transferability.** Evaluating the downstream performance of pretrained GNNs under distribution shift is nontrivial, because target-domain test error $\mathrm{err}_{\mathrm{test}}$ is hard to analyze directly. To address that, we adopt Fisher separability in target domain $F_T := \min_{c \neq c'} \|\mu_T^c - \mu_T^{c'}\|_2^2 / \max_k \mathrm{Tr}(\Sigma_T^k)$, where $\mu_T^c$ and $\Sigma_T^c$ are the class-wise mean and covariance of the output representation (definition 1).

*Theorem* 1. For a linear probe trained on $n$ labeled target nodes, there exist explicit constants $c_{\mathrm{l}}, c_{\mathrm{u}} > 0$ such that

$$\frac{c_{\mathrm{l}}}{\sqrt{n\,F_T}} \;\le\; \mathbb{E}\big[\mathrm{err}_{\mathrm{test}}\big] \;\le\; \frac{c_{\mathrm{u}}}{\sqrt{n\,F_T}}, \quad \text{and} \quad \mathrm{err}_{\mathrm{test}} \;\le\; \frac{c_{\mathrm{u}}}{\sqrt{n\,F_T}} + \sqrt{\frac{\log(1/\delta)}{2n}} \;\; (\text{w.p. } 1-\delta).$$

Equivalently, to achieve $\mathbb{E}[\mathrm{err}_{\mathrm{test}}] \le \varepsilon$ it suffices that labeled node number $n \ge c_{\mathrm{u}}^2/(F_T\,\varepsilon^2)$.

Theorem 1 establishes that expected test error and $1/\sqrt{nF_T}$ are of the same order of magnitude, with high probability guarantees. Therefore, a larger $F_T$ implies lower test error and sample complexity. This also makes $F_T$ a reliable surrogate for analyzing the target performance. Next, under the context of graph pre-training, we further decompose $F_T$ into two terms as factors that determine the lower bound of $F_T$.

*Proposition* 1. Under a bounded within-class scatter $\sigma_{\max}^2 = \max_k \mathrm{Tr}(\Sigma_T^k)$, we have

$$\sqrt{F_T} \;\ge\; \frac{1}{\sigma_{\max}} \Big( \underbrace{\mathbb{E}_{i \neq k}\big\|\mu_S^i - \mu_S^k\big\|}_{\text{source separation (architecture-driven)}} \;-\; \underbrace{\mathbb{E}_{i,j}\big\|\mu_T^j - \mu_S^i\big\|}_{\text{source–target alignment (shift)}} \Big).$$

For a $L$-layer encoder, we note $\Delta_S(L) := \mathbb{E}_{i \neq k}\big\|\mu_S^i(L) - \mu_S^k(L)\big\|$, $\Delta_{ST}(L) := \mathbb{E}_{i,j}\big\|\mu_T^j(L) - \mu_S^i(L)\big\|$. The first term measures how well the pretrained encoder separates representations from different class in the source domain, reflecting the pretraining capability, while the second quantifies the distance between source and target representations which corresponds to the distribution shifts.

**(I) Architecture–driven source separation** $\Delta_S(L) := \mathbb{E}_{i \neq k}\big\|\mu_S^i(L) - \mu_S^k(L)\big\|$ :

*Theorem* 2. Let $\theta_{\mathrm{arch}}(d)$ denote constant function with $d$, and $C_{\mathrm{noise}}$ denote the noise coefficient. Then for any depth $L \ge 1$,

$$\Delta_S(L) \;\ge\; \underbrace{a^L}_{\text{depth}} \Delta_S(0) - \underbrace{\frac{C_{\mathrm{noise}}}{\sqrt{N_{\mathrm{pretrain}}}}}_{\text{data size}} \underbrace{\frac{a^L - 1}{a - 1}}_{\text{depth accumulation}}, \qquad a = 1 + \underbrace{\theta_{\mathrm{arch}}(d)}_{\text{architecture \& width}} \qquad (1)$$

the bound increases with depth $L$ and width $d$ (via $\theta_{\mathrm{arch}}(d)$), and increases with data size $N_{\mathrm{pretrain}}$ (the subtractive term shrinks as $N_{\mathrm{pretrain}}^{-1/2}$).

Theorem 2 shows $\Delta_S(L)$ is jointly governed by four factors: (i) the initial class-mean separation in the data $\Delta_S(0)$(e.g., the prototype norm $\sigma$ in Shift 2), (ii) architecture & width $\theta_{\mathrm{arch}}(d)$, (iii) the pretraining size $N_{\mathrm{pretrain}}$, and (iv) inherent model width $d$ and depth $L$.

In particular, enlarging width and depth increases an lower bound on $\mathbb{E}_{i,k}\|\mu_S^i - \mu_S^k\|$, aligning with the positive model-scaling trend in Sec. 3.1. Moreover, as stated in corollary 7, due to learnable aggregation, under some conditions, graph transformers can lead to a more favorable $\theta_{\mathrm{arch}}(d)$, which enlarges $\Delta_S(L)$, and aligns with the observations in Sec. 3.4–3.5 that graph transformer demonstrates superior scaling effect.

**(II) Source–target alignment** $\Delta_{ST}(L) := \mathbb{E}_{i,j}\big\|\mu_T^j(L) - \mu_S^i(L)\big\|$**:** To quantify the shift between source and target domains, we adopt the class-conditional Wasserstein distance as a metric of distribution shift. Let $\varepsilon_A$ and $\varepsilon_X$ denote the Wasserstein distances over structural and feature distributions, respectively:

*Theorem* 3. Let the single architecture factor $\phi_{\text{arch}}(d) \in (0, \sqrt{d})$ collect all architecture-dependent constants (activation Lipschitz, and propagation gap). Define $r(d) := \frac{\phi_{\text{arch}}(d)}{\sqrt{d}} \in (0, 1)$. Then for any depth $L \geq 1$,

$$-\Delta_{ST}(L) \geq - \left( \underbrace{r(d)^L}_{\textbf{depth}} \epsilon_X + \underbrace{\frac{r(d)}{\sqrt{n_{\min}}}}_{r(d) \sim (d^{-1/2})} \cdot \underbrace{\frac{1 - r(d)^L}{1 - r(d)}}_{\textbf{depth accumulation}} \epsilon_A \right) \tag{2}$$

The bound in Theorem 3 is shaped by two opposing factors: larger model complexity with higher depth and width tightens it, thus improving source–target alignment and the target Fisher separability $F_T$; whereas larger distribution shifts in $\varepsilon_A$ or $\varepsilon_X$ loosens it. This trade-off explains Sec. 3.3, where positive data scaling under severe shifts (e.g., Shift 5) emerges only with sufficient capacity.

## 5 CONCLUSION

This work provides the first systematic investigation of scaling laws in graph self-supervised pre-training under distribution shifts. Our key finding is that increasing model capacity consistently improves transferability, while the effectiveness of data scaling is conditional on model expressiveness and architecture; most notably with graph transformers. This rule holds across both synthetic benchmarks with controlled shifts and real-world datasets. We further validate and explain these dynamics through a theoretical framework that disentangles the roles of model architecture, capacity, and distribution shift. In total, our results offer concrete principles on model architecture and dataset features to enable design of scalable and robust graph foundation models.

## 6 FUTURE DIRECTIONS

In this work, we focus on node-level classification under distribution shifts, while extending our scaling analysis to graph-level tasks, like molecular property prediction, is an important direction for future work. We prioritize the node-level setting because its scaling behavior remains poorly understood: prior studies report inconsistent or even negative returns from scaling data and models, in contrast to the more intuitive scaling often observed on graph-level benchmarks. A node-centric formulation also allows us to isolate how homophily changes, feature noise, and other local structural shifts affect transfer, without additional confounding factors introduced by graph-level pooling. Since node- and graph-level tasks rely on different inductive biases, which one focusing on local neighborhood aggregation and separation while the other involves global structural isomorphism and pooling, we expect that a unified treatment will require tailored analyses and benchmark design, which we leave to future work.

## 7 LLM USAGE DISCLOSURE

We use LLMs solely as writing-assist tools to polish the manuscript. All research ideas, methodology, experiments, theoretical analyses, and initial drafts were conceived and written by the authors. LLMs were only used to refine grammar, improve clarity, and enhance readability of text already written by the authors.

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

# A  THEORETICAL ANALYSIS: LIMITS AND DRIVERS OF PRETRAINING TRANSFERABILITY

In this section, we present a theoretical analysis of graph pretraining under distribution shift. We develop a unified framework that combines Fisher separability and Wasserstein domain divergence to characterize the discriminability and robustness of pretrained representations. We show that the Fisher score governs both the generalization error and sample complexity of downstream linear probing. We then analyze how structural and feature shifts propagate through graph networks, ultimately impacting transfer performance. Based on these insights, we derive a unified upper bound on test-time accuracy, highlighting how model capacity, architectural choices, and domain differences jointly determine the scalability of graph pretraining.

## A.1  PROBLEM SETUP: WHAT LIMITS PRETRAINING TRANSFER?

We aim to theoretically understand the factors that determine the success or failure of graph pretraining under distribution shift. To this end, we consider a standard transfer learning setup, where a node encoder $f_\theta$ is first pretrained on an unlabeled source graph $\mathcal{G}_S = (A_S, X_S)$, and then evaluated on a labeled target graph $\mathcal{G}_T = (A_T, X_T, Y_T)$ via few-shot linear probing.

We assume $f_\theta$ is an $L$-layer GNN or graph transformer with hidden dimension $d$, which generates node representations across layers. For each node $v$ in domain $D \in S, T$, we denote the layer-$\ell$ embedding by $H_D^v(\ell) \in \mathbb{R}^d$, where $\ell = 0$ corresponds to the input feature $X_D^v$. The final-layer embedding $H_D^v(L)$ is used for downstream classification.

To capture distributional differences, we define the class-conditional mean embeddings in domain $D$ as $\mu_D^c(\ell) = \mathbb{E}[H_D^v(\ell) \mid Y_D^v = c]$, and the within-class covariance in the target domain as $\Sigma_T^c = \text{Cov}(H_T^v(L) \mid Y_T^v = c)$. We also define $\mu_D^i(\ell)$ as the class-mean embedding at layer $\ell$ for the class associated with node $i$ in domain $D$. These notations allow us to systematically analyze how embeddings evolve across layers, domains, nodes, and classes.

Our central question is: what intrinsic properties of the learned embedding space in Graph Pre-training determine downstream generalization accuracy under domain shift?

**Fisher Separability as a Transferability Metric.**    To quantify how well the pretrained embedding space separates downstream classes, we introduce the Fisher separability score (Fisher, 1936) , a classic geometric criterion widely used in representation learning and classification task.

*Definition* 1 (Fisher separability).  Fix a domain $\mathcal{D} \in \{S, T\}$. Let $\{H_{\mathcal{D}}^v\}_{v=1}^{n_{\mathcal{D}}} \subset \mathbb{R}^d$ be the final-layer node embeddings produced by the pretrained encoder $f_\theta$ on domain $\mathcal{D}$, and let $y_{\mathcal{D}}^v \in \{1, \ldots, C\}$ be the class label of node $v$. For each class $c$, define the class-$c$ node set

$$S_{\mathcal{D}}^c \; := \; \{\, v \; : \; y_{\mathcal{D}}^v = c \,\},$$

the class mean embedding

$$\mu_{\mathcal{D}}^c \; := \; \frac{1}{|S_{\mathcal{D}}^c|} \sum_{v \in S_{\mathcal{D}}^c} H_{\mathcal{D}}^v,$$

and the within-class covariance

$$\Sigma_{\mathcal{D}}^c \; := \; \frac{1}{|S_{\mathcal{D}}^c|} \sum_{v \in S_{\mathcal{D}}^c} \left( H_{\mathcal{D}}^v - \mu_{\mathcal{D}}^c \right)\left( H_{\mathcal{D}}^v - \mu_{\mathcal{D}}^c \right)^\top.$$

The (domain-indexed) Fisher separability score is

$$F_{\mathcal{D}} \; := \; \frac{\min\limits_{c \neq c'} \left\| \mu_{\mathcal{D}}^c - \mu_{\mathcal{D}}^{c'} \right\|_2^2}{\max\limits_{k} \mathrm{Tr}\big(\Sigma_{\mathcal{D}}^k\big)} \, .$$

Intuitively, the numerator measures the minimal squared distance between class centroids, while the denominator captures the maximum intra-class variance. A higher $\mathcal{F}$ implies that class clusters are more tightly packed and well-separated—favorable conditions for node classification with few labeled samples.

**Why Fisher Score Matters .**    In the few-shot transfer setting, the success of downstream linear probing depends heavily on how well the class structure is preserved in the embedding space learned during self-supervised pretraining. Thus, Fisher separability provides a tractable and interpretable surrogate for analyzing pretraining effectiveness. (Gu et al., 2012; Dixit et al., 2019). In the next subsection, we formally show that $\mathcal{F}$ controls both the sample complexity and generalization error of linear classifier.

### A.2    GENERALIZATION BOUNDS GOVERNED BY FISHER SCORE.

*Assumption* 1 (IID labeled samples and class prior lower bound).  The target domain provides $n$ i.i.d. labeled samples $(x, y) \sim \mathcal{D}_T$ over classes $\{1, \ldots, C\}$ with priors $\pi_c \geq \pi_{\min} > 0$.

*Assumption* 2 (Bounded intra-class variance (target)).  For each class $c$, the within-class covariance of target embeddings satisfies $\mathrm{Tr}(\Sigma_T^c) \leq \sigma^2$ for some finite $\sigma > 0$.

*Assumption* 3 (Separated class means).  For all $c \neq c'$, $\|\mu_T^c - \mu_T^{c'}\|^2 \geq D^2$ for some $D > 0$. Consequently, the Fisher score $F_T := \frac{\min_{c \neq c'} \|\mu_T^c - \mu_T^{c'}\|^2}{\max_k \mathrm{Tr}(\Sigma_T^k)}$ satisfies $F_T \geq D^2/\sigma^2$.

*Assumption* 4 (Frozen representation + linear probe and mild tails).  A linear classifier (probe) is trained on frozen target embeddings to zero training error (with optional $\ell_2$ regularization and feature $\ell_2$-normalization). The class-conditionals are sub-Gaussian with parameter $\nu^2$ (or have bounded radius), ensuring standard margin generalization bounds with explicit constants.

*Theorem* 4 (Fisher-controlled transfer generalization: two-sided rate).  Suppose Assumptions 1–4 hold.  Then there exist positive constants $c_{\mathrm{u}}, c_{\mathrm{l}}$ depending only on $(\sigma, \nu, \pi_{\min})$ and the probe's normalization/regularization scheme such that

$$\frac{c_{\mathrm{l}}}{\sqrt{n\, F_T}} \;\; \leq \;\; \mathbb{E}\big[\mathrm{err}_{\mathrm{test}}\big] \;\; \leq \;\; \frac{c_{\mathrm{u}}}{\sqrt{n\, F_T}}. \tag{3}$$

Moreover, the following high-probability upper bound holds for any $\delta \in (0, 1)$:

$$\mathrm{err}_{\mathrm{test}} \;\; \leq \;\; \frac{c_{\mathrm{u}}}{\sqrt{n\, F_T}} + \sqrt{\frac{\log(1/\delta)}{2n}} \qquad \text{with probability at least } 1 - \delta. \tag{4}$$

*Proof sketch.* (Upper bound) Under 4, margin-based bounds for linear classifiers yield $\text{err}_{\text{test}} \leq \tilde{c}(\sigma, \nu, \pi_{\min})/(\gamma\sqrt{n})$ with an explicit constant $\tilde{c}$. Using 2–3 one obtains $\gamma \gtrsim \min_{c \neq c'} \|\mu_T^c - \mu_T^{c'}\|/\sigma = \sqrt{F_T}$, leading to the RHS of equation 3; a standard Chernoff/Hoeffding step gives equation 4. (Lower bound) Construct a $C$-ary hypothesis family by placing class means at least $D$ apart and injecting within-class noise with covariance trace $\sigma^2$. A Fano/packing argument yields $\mathbb{E}[\text{err}_{\text{test}}] \geq c_l/(\sqrt{n\,F_T})$ with $c_l$ depending on $(\sigma, \nu, \pi_{\min})$. Combining gives equation 3. $\qquad\square$

*Corollary* 5 (Sample complexity). Under the conditions of thm:fisher-two-sided, to guarantee $\mathbb{E}[\text{err}_{\text{test}}] \leq \varepsilon$ it suffices that

$$n \;\geq\; \frac{c_{\text{u}}^2}{F_T\,\varepsilon^2}.$$

Conversely, any procedure achieving $\mathbb{E}[\text{err}_{\text{test}}] \leq \varepsilon$ must have $n \geq c/\left(F_T\,\varepsilon^2\right)$ for a universal constant $c > 0$ (depends on $(\sigma, \nu, \pi_{\min})$ only).

*Remark* 1. This result highlights the dual importance of $\mathcal{F}$: higher Fisher scores reduce generalization error and decrease the number of labeled examples required.

### A.3 Proof details for Theorem 4

We restate that

$$F_T \;:=\; \frac{\min\limits_{c \neq c'} \left\|\mu_T^c - \mu_T^{c'}\right\|^2}{\max\limits_{k} \text{Tr}\left(\Sigma_T^k\right)}$$

and Assumptions 1–4 hold. We proceed through three lemmas.

*Lemma* 1 (From Fisher separability to margin). Suppose Assumptions 2–4. Let features be $\ell_2$-normalized (or the classifier is $\ell_2$-regularized so that $\|w\|_2 \leq 1$ w.l.o.g.). There exists a universal constant $c_\gamma > 0$ depending only on the tail proxy ($\nu$ or radius) such that, with probability at least $1 - \delta$ over the draw of the $n$ labeled target samples,

$$\gamma \;\geq\; c_\gamma\, \frac{\min\limits_{c \neq c'} \left\|\mu_T^c - \mu_T^{c'}\right\|}{\sqrt{\max\limits_{k} \text{Tr}\left(\Sigma_T^k\right)}} \;-\; r_n(\delta), \qquad r_n(\delta) \;=\; O\!\left(\sqrt{\frac{\log(C/\delta)}{n}}\right).$$

Consequently, ignoring the vanishing $r_n(\delta)$ term for readability, $\gamma \gtrsim \sqrt{F_T}$ up to the constant $c_\gamma$.

*Proof.* (1) (Class means concentration) By sub-Gaussian (or bounded-radius) assumption, $\|\widehat{\mu}_T^c - \mu_T^c\| = O\!\left(\sqrt{1/n_T^c}\right)$ uniformly over $c$ with high probability. Similarly, classwise covariances concentrate around $\Sigma_T^c$ so that empirical within-class radii are $O\!\left(\sqrt{\text{Tr}(\Sigma_T^c)}\right)$.

(2) (Geometric separability) For any pair $(c, c')$, by triangle inequality

$$\min_{x \in \text{class } c,\; x' \in \text{class } c'} \|x - x'\| \;\geq\; \left\|\mu_T^c - \mu_T^{c'}\right\| \;-\; R_T^c \;-\; R_T^{c'},$$

where $R_T^c$ bounds the empirical dispersion of class-$c$ target samples by $O\!\left(\sqrt{\text{Tr}(\Sigma_T^c)}\right)$ with high probability.

(3) (Margin lower bound) A linear separator aligned with the vector connecting empirical means achieves margin at least

$$\frac{\left\|\mu_T^c - \mu_T^{c'}\right\| - R_T^c - R_T^{c'}}{\|w\|_2\,\sigma_{\max}},$$

where $\sigma_{\max}^2 = \max_k \text{Tr}(\Sigma_T^k)$. With feature or weight normalization and a union bound over all class pairs, we get the stated inequality with $r_n(\delta)$ absorbing sampling fluctuations and the constant $c_\gamma$ coming from norm conventions. $\qquad\square$

*Lemma* 2 (Linear classifier margin bound). Under Assumptions 1 and 4, for a linear classifier achieving empirical margin $\gamma > 0$ on $n$ labeled target examples (features normalized; weights with

bounded norm), there exists $c_{\mathrm{u}} > 0$ depending on $(\sigma, \nu, \pi_{\min})$ and the normalization/regularization scheme such that

$$\mathbb{E}\big[\mathrm{err}_{\mathrm{test}}\big] \ \leq \ \frac{c_{\mathrm{u}}}{\gamma\sqrt{n}}.$$

Moreover, for any $\delta \in (0, 1)$,

$$\mathrm{err}_{\mathrm{test}} \ \leq \ \frac{c_{\mathrm{u}}}{\gamma\sqrt{n}} \ + \ \sqrt{\frac{\log(1/\delta)}{2n}} \quad \text{with probability at least } 1 - \delta.$$

*Proof.* Use a standard margin/Rademacher-complexity argument for linear separators with bounded norm under $\ell_2$-normalized inputs: the empirical margin controls the surrogate loss (e.g., hinge), yielding generalization in terms of $\mathfrak{R}_n(\mathcal{H}) \asymp 1/\sqrt{n}$ multiplied by $1/\gamma$. The constant depends on tails ($\nu$ or radius), label prior lower bound $\pi_{\min}$ (to avoid degenerate classes), and the normalization scheme. A Chernoff/Hoeffding step converts surrogate excess risk to 0–1 error. $\square$

*Lemma* 3 (Fano/Le Cam lower bound). Under Assumptions 1–3 (and tails in 4), there exists $c_{\mathrm{l}} > 0$ depending on $(\sigma, \nu, \pi_{\min})$ such that for any classifier trained on $n$ labeled target samples,

$$\mathbb{E}\big[\mathrm{err}_{\mathrm{test}}\big] \ \geq \ \frac{c_{\mathrm{l}}}{\sqrt{n\, F_T}}.$$

*Proof.* Construct a finite hypothesis family indexed by class means so that any two hypotheses differ by at least $D$ in their mean configuration while sharing the same within-class covariance (trace $\leq \sigma^2$). For mixtures with equal priors, the per-sample KL divergence between two hypotheses is bounded by $O\big(D^2/\sigma^2\big) = O(F_T)$. By independence, the $n$-sample KL is $O(nF_T)$. Apply Le Cam's method (two-point testing) or Fano's inequality over a packing of hypotheses to show that the average misclassification probability cannot be smaller than a constant times $1/\sqrt{nF_T}$; the square-root rate arises by inverting the typical inequality $\alpha + \beta \geq 1 - \sqrt{\frac{1}{2}\mathrm{KL}}$ and optimizing constants. $\square$

**Putting together.** From Lemma 1, with high probability $\gamma \gtrsim \sqrt{F_T}$ up to $c_\gamma$ and a negligible $r_n(\delta)$. Plugging into Lemma 2 gives

$$\mathbb{E}\big[\mathrm{err}_{\mathrm{test}}\big] \ \leq \ \frac{c_{\mathrm{u}}}{\gamma\sqrt{n}} \ \lesssim \ \frac{c_{\mathrm{u}}}{c_\gamma} \cdot \frac{1}{\sqrt{n\, F_T}},$$

which is the desired upper bound in Theorem 4 (up to replacing $c_{\mathrm{u}}/c_\gamma$ by a single constant). The lower bound follows directly from Lemma 3. This proves the two-sided rate equation 3 and the high-probability bound equation 4. $\square$

## A.4 Additive Upper Bound on Fisher Score

For domain $\mathcal{D} \in \{S, T\}$ and class $c \in \{1, \dots, C\}$, let $\mu_{\mathcal{D}}^c \in \mathbb{R}^d$ be the class-$c$ mean embedding and $\Sigma_{\mathcal{D}}^c \in \mathbb{R}^{d \times d}$ the within-class covariance of the final-layer embeddings. Define

$$\sigma_{\max}^2 \ := \ \max_k \mathrm{Tr}\big(\Sigma_T^k\big), \qquad F_T \ := \ \frac{\min\limits_{c \neq c'} \big\|\mu_T^c - \mu_T^{c'}\big\|_2^2}{\max\limits_k \mathrm{Tr}\big(\Sigma_T^k\big)}.$$

We use $\mathbb{E}_{i \neq k}$ for the uniform average over unordered class pairs $\{(i, k) : i \neq k\}$, and $\mathbb{E}_{i,j}$ for the uniform average over all $(i, j)$.

*Assumption* 5 (Bounded within-class scatter in the target domain). $\max\limits_k \mathrm{Tr}\big(\Sigma_T^k\big) \leq \sigma_{\max}^2 < \infty$.

*Assumption* 6 (Optional concentration (for a refined remark)). Class-mean estimators are sub-Gaussian (or have bounded radius), so that for all $c$ the empirical mean $\widehat{\mu}_T^c$ concentrates around $\mu_T^c$ and order-statistics of pairwise distances admit standard concentration bounds.

*Proposition* 2 (Additive upper bound on $\sqrt{F_T}$ with explicit constants). Under Assumption 5,

$$\sqrt{F_T} \ \geq \ \frac{1}{\sigma_{\max}} \Big( \underbrace{\mathbb{E}_{i \neq k} \|\mu_S^i - \mu_S^k\|}_{\text{source separation}} - \underbrace{\mathbb{E}_{i,j} \|\mu_T^j - \mu_S^i\|}_{\text{source–target alignment}} \Big).$$

*Proof sketch.* Replace $\min$ by an expectation .

$$F_T \;=\; \frac{\min_{c\neq c'}\|\mu_T^c - \mu_T^{c'}\|^2}{\sigma_{\max}^2} \;\leq\; \frac{\mathbb{E}_{c\neq c'}\|\mu_T^c - \mu_T^{c'}\|^2}{\sigma_{\max}^2}.$$

For any classes $(j_1, j_2)$ and any source indices $(i, k)$,

$$\|\mu_T^{j_1} - \mu_T^{j_2}\| \;\geq\; \|\mu_S^i - \mu_S^k\| - \|\mu_T^{j_1} - \mu_S^i\| - \|\mu_T^{j_2} - \mu_S^k\|.$$

This yields

$$\sqrt{F_T} \;\geq\; \frac{1}{\sigma_{\max}}\Big(\mathbb{E}_{i\neq k}\|\mu_S^i - \mu_S^k\| - 2\cdot\mathbb{E}_{i,j}\|\mu_T^j - \mu_S^i\|\Big),$$

which is the claim. $\qquad\square$

## A.5   Source separation

*Definition* 2 (Layer-wise source separation).  For layer $\ell \in \{0, 1, \dots\}$, denote the source-domain class means by $\mu_S^i(\ell) \in \mathbb{R}^d$ and define

$$\Delta_S(\ell) \;:=\; \mathbb{E}_{i\neq k}\left\|\mu_S^i(\ell) - \mu_S^k(\ell)\right\|.$$

Let $\pi$ be the stationary distribution of the propagation operator $P^{(\ell)}$ and set $M := I - \mathbf{1}\pi^\top$. Then class-mean differences lie in the subspace $\mathcal{V}_{\mathrm{diff}} := \mathrm{Im}(M) = \{x \in \mathbb{R}^d : \pi^\top x = 0\}$.

*Assumption* 7 ( GNN block and lower-Lipschitz nonlinearity).  Each layer is a aggregation block

$$H^{(\ell+1)} \;=\; H^{(\ell)} \;+\; \sigma\big(P^{(\ell)}H^{(\ell)}W^{(\ell)}\big),$$

where the activation $\sigma$ is lower-Lipschitz with slope $m \in (0, 1]$ on the support of representations:

$$\|\sigma(u) - \sigma(v)\| \;\geq\; m\,\|u - v\| \qquad (\forall\, u, v).$$

*Assumption* 8 (Architecture-dependent non-degeneracy on $\mathcal{V}_{\mathrm{diff}}$).  Let $\rho_\ell := \sigma_{\min}\big(MP^{(\ell)}M\big)$ and $\omega_\ell := \sigma_{\min}\big(W^{(\ell)}\big)$. Assume uniform lower bounds

$$\rho_\ell \;\geq\; \underline{\rho}_{\mathrm{arch}} \;>\; 0, \qquad \omega_\ell \;\geq\; \underline{\omega}(d) \;>\; 0,$$

where the architecture-dependent constant $\underline{\rho}_{\mathrm{arch}}$ is

$$\underline{\rho}_{\mathrm{arch}} = \begin{cases} \sigma_{\min}(MPM), & \text{GCN / MP-GNN,} \\ (1-\tau)\,\sigma_{\min}(M\widetilde{A}\,M), & \text{Graph Transformer with } A = (1-\tau)\widetilde{A} + \tau\,\mathbf{1}\pi^\top. \end{cases}$$

(Here $M(\mathbf{1}\pi^\top)M = 0$, so only $\widetilde{A}$ contributes on $\mathcal{V}_{\mathrm{diff}}$.)

*Assumption* 9 (Width scaling and data noise).  The network width $d$ controls the minimal linear gain via a nondecreasing function $\underline{\omega}(d)$ (e.g., spectral normalization can yield $\underline{\omega}(d) \propto \sqrt{d}/\kappa_W(d)$). For per-class sample size $N_{\mathrm{pretrain}}$, the class-mean estimation noise satisfies

$$\mathbb{E}\big\|\mathrm{perturb}_\ell\big\| \;\leq\; \frac{C_{\mathrm{noise}}(d)}{\sqrt{N_{\mathrm{pretrain}}}} \qquad (\ell = 0, 1, \dots, L-1),$$

where $C_{\mathrm{noise}}(d)$ is nondecreasing in $d$ (absorbing post-layer Lipschitz factors).

*Lemma* 4.  Under Assumptions 7–9, for any unordered class pair $(i, k)$,

$$\big\|\mu_S^i(\ell+1) - \mu_S^k(\ell+1)\big\| \;\geq\; \Big(1 + m\,\rho_\ell\,\omega_\ell\Big)\big\|\mu_S^i(\ell) - \mu_S^k(\ell)\big\| \;-\; 2\,\|\mathrm{perturb}_\ell\|.$$

Averaging over $(i, k)$ yields

$$\Delta_S(\ell+1) \;\geq\; a_\ell\,\Delta_S(\ell) \;-\; \frac{C_{\mathrm{noise}}(d)}{\sqrt{N_{\mathrm{pretrain}}}}, \qquad a_\ell := 1 + m\,\rho_\ell\,\omega_\ell.$$

*Proposition* 3 (Uniform-gain recursion). If $a_\ell \geq a_\star := 1 + m\,\underline{\rho}_{\text{arch}}\,\underline{\omega}(d) > 1$ for all $\ell$, then

$$\Delta_S(\ell+1) \;\geq\; a_\star\,\Delta_S(\ell) \;-\; \frac{C_{\text{noise}}(d)}{\sqrt{N_{\text{pretrain}}}} \qquad (\ell = 0, \ldots, L-1).$$

Solving the linear recursion gives, for any $L \geq 1$,

$$\Delta_S(L) \;\geq\; a_\star^L\,\Delta_S(0) \;-\; \frac{C_{\text{noise}}(d)}{\sqrt{N_{\text{pretrain}}}}\,\frac{a_\star^L - 1}{a_\star - 1}.$$

*Theorem* 6. Under Assumptions 7–9, define the architecture–width gain

$$\eta_{\text{arch}}(d) \;:=\; m\,\underline{\rho}_{\text{arch}}\,\underline{\omega}(d) \quad (> 0), \qquad a_\star \;:=\; 1 + \eta_{\text{arch}}(d).$$

Then for any depth $L \geq 1$,

$$\boxed{\;\Delta_S(L) \;\geq\; \big(1 + \eta_{\text{arch}}(d)\big)^L\,\Delta_S(0) \;-\; \frac{C_{\text{noise}}(d)}{\sqrt{N_{\text{pretrain}}}} \cdot \frac{\big(1 + \eta_{\text{arch}}(d)\big)^L - 1}{\eta_{\text{arch}}(d)}\;.\;} \tag{5}$$

The bound increases with depth $L$ (since $a_\star > 1$); increases with width $d$ whenever $\underline{\omega}(d)$ is nondecreasing; and increases with data size $N_{\text{pretrain}}$ because the subtractive term scales as $N_{\text{pretrain}}^{-1/2}$.

*Proof sketch.* By the residual update, $\Delta_{\ell+1}^{ik} = \Delta_\ell^{ik} + \sigma(P^{(\ell)}\Delta_\ell^{ik}W^{(\ell)}) + \text{noise}$. Using the reverse triangle inequality, the lower-Lipschitz property of $\sigma$, and the product lower bound for smallest singular values on $\mathcal{V}_{\text{diff}}$ gives $\|\sigma(P\Delta W)\| \geq m\,\sigma_{\min}(MPM)\sigma_{\min}(W)\|\Delta\|$. This yields Lemma 4. Apply $\rho_\ell \geq \underline{\rho}_{\text{arch}}$ and $\omega_\ell \geq \underline{\omega}(d)$ to obtain $a_\star$.

Solving $x_{\ell+1} \geq a_\star x_\ell - b$ with $b = C_{\text{noise}}(d)/\sqrt{N_{\text{pretrain}}}$ yields equation 5. Monotonicity then follows directly from equation 5. $\qquad\square$

*Corollary* 7 (GCN vs. Graph Transformer (architecture term)). Fix width $d$ and data size $N_{\text{pretrain}}$. The lower bound for a Graph Transformer dominates that of a GCN iff

$$(1 - \tau)\,\sigma_{\min}\big(M\widetilde{A}M\big) \;\geq\; \sigma_{\min}\big(MPM\big),$$

i.e., the attention conditioning gain offsets the teleport factor $(1 - \tau)$. Equivalently, let $g := \frac{\sigma_{\min}(M\widetilde{A}M)}{\sigma_{\min}(MPM)}$ and $t := 1 - \tau$; then the GT bound $\geq$ the GCN bound iff $t\,g \geq 1$ (strictly larger when $tg > 1$).

## A.6 TRANSFER GENERALIZATION UNDER DOMAIN SHIFT

We analyze how source→target distribution shift propagates through $L$-layer GNNs and yields a target-side class-mean discrepancy bound that decreases with depth $L$, width $d$, and data size $n$.

**Wasserstein distances and shift magnitudes**

*Definition* 3 (2-Wasserstein distance). Let $P, Q$ be probability measures on a metric space $(\mathcal{X}, \|\cdot\|_2)$.

$$W_2(P, Q) \;:=\; \Big(\inf_{\gamma \in \Pi(P,Q)} \mathbb{E}_{(x,y)\sim\gamma}\|x - y\|_2^2\Big)^{1/2},$$

where $\Pi(P, Q)$ is the set of couplings of $P$ and $Q$.

We write the feature and adjacency shifts as

$$\varepsilon_X := W_2(P_T^X, P_S^X), \qquad \varepsilon_A := W_2(P_T^A, P_S^A),$$

and assume classwise target scatter is bounded: $\sigma_{\max}^2 := \max_k \text{Tr}(\Sigma_T^k) < \infty$.

**Layerwise shift propagation (GCN case)** Consider an $L$-layer GCN with updates $H^{(\ell+1)} = \sigma\big(\widehat{A}\, H^{(\ell)} W^{(\ell)}\big)$, $H^{(0)} = X$, where $\widehat{A}$ is symmetric normalized adjacency and $\sigma$ is $\kappa$-Lipschitz. Let

$$\lambda := 1 - \max\{|\lambda_2(\widehat{A})|, |\lambda_n(\widehat{A})|\} \in [0, 1]$$

be the (uniform) spectral gap, and define the layer-$\ell$ cross-domain class-mean discrepancy $\Delta^{(\ell)} := \mathbb{E}_{i,j}\|\mu_T^j(\ell) - \mu_S^i(\ell)\|$. Assume the one-layer map is Lipschitz in $(X, A)$ with constants $C_X, C_A > 0$.

*Assumption* 10 (Width normalization). There exists $B_0 > 0$ such that each linear layer obeys $\|W^{(\ell)}\|_2 \le \frac{B_0}{\sqrt{d}}$ ($\ell = 0, \dots, L-1$), e.g., via spectral normalization/weight normalization. Set $C := \kappa$.

*Lemma* 5 (Layerwise shift recursion with $d$-explicit contraction). Under the above setting and Assumption 10, with

$$r(d) := (CB_0)\,(1 - \lambda)\,\frac{1}{\sqrt{d}} \in (0, 1),$$

the layerwise shift satisfies, for all $L \ge 1$,

$$\Delta^{(L)} \le r(d)^L\, \varepsilon_X + \frac{r(d)}{1 - r(d)}\big(1 - r(d)^L\big)\, C_A\, \varepsilon_A. \tag{6}$$

*Proof sketch.* We have $\Delta^{(\ell+1)} \le (CB)\,(1 - \lambda)\,\Delta^{(\ell)} + (CB)\, C_A\, \varepsilon_A$ with $B = \|W^{(\ell)}\|_2$. Using Assumption 10 gives $B \le B_0/\sqrt{d}$, hence the contraction factor $r(d) = (CB_0)(1-\lambda)/\sqrt{d}$. Unrolling the linear recursion and using $\Delta^{(0)} \le \varepsilon_X$ yields equation 6. $\square$

*Theorem* 8. Let $r(d)$ be as in Lemma 5 and define the aggregated shift $\varepsilon_{\text{shift}} := C_X \varepsilon_X + C_A \varepsilon_A$. Then for any $L \ge 1$,

$$\boxed{\Delta^{(L)} \le r(d)^L\, \varepsilon_X + \frac{r(d)}{1 - r(d)}\big(1 - r(d)^L\big)\, C_A\, \varepsilon_A, \qquad r(d) = \frac{(CB_0)(1 - \lambda)}{\sqrt{d}}.} \tag{7}$$

In particular, if $r(d) \le \frac{1}{2}$,

$$\Delta^{(L)} \le r(d)^L\, \varepsilon_X + \frac{2\,(CB_0)(1 - \lambda)}{\sqrt{d}}\, C_A\, \varepsilon_A.$$

Moreover, if the shifts concentrate with data as $\varepsilon_{\text{shift}} \le \frac{C_{\text{noise}}(d)}{\sqrt{n}}$, then

$$\Delta^{(L)} \le r(d)^L\, \frac{C_{\text{noise}}(d)}{\sqrt{n}} + \frac{r(d)}{1 - r(d)}\big(1 - r(d)^L\big)\, \frac{C_{\text{noise}}(d)}{\sqrt{n}}, \tag{8}$$

so the bound decreases with larger $L$, larger $d$ (since $r(d) \propto d^{-1/2}$), and larger $n$.

*Corollary* 9 (Architecture-dependent comparison (GCN vs. Graph Transformer)). For a Graph Transformer with attention kernel decomposition $A = (1 - \tau)\,\widetilde{A} + \tau\, \mathbf{1}\pi^\top$, the contraction reads

$$r_{\text{GT}}(d) = \frac{(CB_0)}{\sqrt{d}}\big(1 - \lambda_{\text{eff}}\big), \qquad 1 - \lambda_{\text{eff}} \le (1 - \tau)\big(1 - \widetilde{\lambda}\big),$$

where $\widetilde{\lambda}$ is the gap of $\widetilde{A}$ on the class-difference subspace ($M(\mathbf{1}\pi^\top)M = 0$). Hence a GT yields a smaller upper bound than GCN whenever

$$(1 - \tau)\big(1 - \widetilde{\lambda}\big) < (1 - \lambda),$$

i.e., attention improves effective contraction beyond the teleport discount.

# B MORE EXPERIMENTAL RESULTS AND DETAILS

## B.1 ANALYSIS OF SCALING CORRELATIONS ACROSS METHODS (FIG.1)

Table 1 compares the Pearson correlation ($R$) between model size and few-shot accuracy for four representative methods. While individual methods differ slightly—with GraphMAE generally showing the strongest correlations, VGAE the weakest, and DGI/GRACE lying in between—the overall trend is consistent across all cases: larger models consistently yield higher downstream accuracy under distribution shifts.

Table 1: Pearson correlation ($R$) between model size (log-parameters) and task accuracy under different distribution shifts. Results are reported for four representative methods: GraphMAE (Hou et al., 2022), VGAE (Kipf & Welling, 2016), DGI (Veličković et al., 2018), and GRACE (You et al., 2020).

| Method | Shift 0 | Shift 1 | Shift 2 | Shift 3 | Shift 4 | Shift 5 |
|---|---|---|---|---|---|---|
| GraphMAE | 0.60 | 0.78 | 0.72 | 0.80 | 0.75 | 0.72 |
| VGAE | 0.08 | 0.47 | 0.46 | 0.51 | 0.44 | 0.43 |
| DGI | 0.51 | 0.66 | 0.61 | 0.67 | 0.65 | 0.62 |
| GRACE | 0.53 | 0.70 | 0.34 | 0.69 | 0.66 | 0.62 |

Table 2: Pearson correlation ($R$) between data size (log-scale) and task accuracy under different distribution shifts. Results are reported for GraphMAE (Hou et al., 2022), VGAE (Kipf & Welling, 2016), DGI (Veličković et al., 2018), and GRACE (You et al., 2020).

| Method | Shift 0 | Shift 1 | Shift 2 | Shift 3 | Shift 4 | Shift 5 |
|---|---|---|---|---|---|---|
| GraphMAE | 0.11 | -0.10 | -0.11 | -0.25 | -0.46 | -0.45 |
| VGAE | 0.16 | -0.04 | -0.07 | -0.22 | -0.42 | -0.42 |
| DGI | 0.13 | -0.05 | -0.10 | -0.24 | -0.43 | -0.42 |
| GRACE | 0.02 | -0.15 | -0.14 | -0.27 | -0.52 | -0.49 |

## B.2   ANALYSIS OF DATA SCALING CORRELATIONS ACROSS METHODS

Table 2 presents the correlations between data size and few-shot accuracy. Here the trend is again consistent across methods: all correlations are close to zero or slightly negative, suggesting that enlarging the pretraining dataset provides little benefit under distribution shifts. The small differences between methods do not alter this conclusion, and the overall results highlight the limited data efficiency of current GNN pretraining approaches.

## B.3   HYPERPARAMETER TABLE OF SSL METHOD

We report the exact hyperparameters for each self-supervised learning (SSL) method used in our study. All configurations are pretrained and evaluated under the same 25-shot linear probing protocol as in Section 3.1; results are averaged over 5 independent random seeds. Unless otherwise noted, Neg ratio denotes the negative-to-positive sampling ratio used in contrastive/edge-based objectives.

## B.4   DETAILED DATA GENERATION DESCRIPTION

We generate each synthetic graph through a modular process consisting of label assignment, feature construction, and structure generation. First, each node is assigned a ground-truth label $y_i \in \{0, 1, \ldots, C - 1\}$ sampled uniformly at random, where the number of classes is fixed at $C = 5$ throughout. Based on these labels, we sample class prototypes $\mu_c \in \mathbb{R}^d$ for each class $c$ uniformly from the unit hypersphere and scale them to a fixed norm. The default norm is $\sigma_1 = 6.0$, which is decreased to $\sigma_2 = 3.0$ in Shift 2 to enhance inter-class separability in the feature space. Importantly, these ground-truth labels $y_i$ are fixed and remain unchanged across all generated graphs; no classifier is involved at any stage. To modulate intra-class ambiguity, we assign each node a soft label distribution $\pi_i \in \Delta^C$, where $\Delta^C$ is the $C$-dimensional probability simplex. In Shifts 0, 2, and 3–5, $\pi_i$ is a one-hot vector determined by $y_i$, while in Shift 1, it is sampled from a softmax distribution, introducing semantic uncertainty. Node features are then constructed as convex combinations of class prototypes, using the soft labels as weights: $x_i = \sum_{c=1}^{C} \pi_{i,c}\mu_c$. We do not apply any extra normalization to input features; the raw generated features (and perturbations) are used directly in both pre-training and linear probing. LayerNorm is applied only inside the GNN encoder for stability, and ablations show that input normalization does not affect scaling trends. This formulation allows independent control over intra-class ambiguity and inter-class separation. After feature generation, graph structure is defined using a contextual stochastic block model (CSBM), where the edge probability between any pair of nodes depends on whether they share the same label. In Shifts 0–2, we set $p_{\text{in}} = 0.02$ and

Table 3: Hyperparameters used for each ssl method

| Method | Hyperparameters |
|--------|-----------------|
| GraphMAE | Node mask rate: 0.5, $\alpha$: 3, Decoder layers: 1, Dropout: 0.2, Learning rate: $1e-3$, Weight decay: 0 |
| LP | Edge batch size: 4096, Neg ratio: 1:1, , Dropout: 0.2, Learning rate: $1e-4$, Weight decay: 0 |
| VGAE | Edge batch size: 4096, Neg ratio: 1:1, Dropout: 0.2, Learning rate: $1e-5$, Weight decay: 0 |
| GRACE | Feature drop: 0.2, Edge drop: 0.2, Dropout: 0.2, Learning rate: $1e-4$, Weight decay: 0 |
| DGI | Edge batch size: 4096, Dropout: 0.2, Learning rate: $1e-5$, Weight decay: 0 |

$p_{\text{out}} = 0.005$ to induce homophily; in Shifts 3 and 5, we reduce homophily by setting $p_{\text{in}} = 0.015$ and $p_{\text{out}} = 0.01$. To further simulate distribution shift in the structural space, we apply edge-level noise in Shifts 4 and 5. Specifically, Shift 4 randomly deletes 10% of existing edges and inserts 5% of new ones between randomly chosen node pairs. Shift 5 combines this corruption with the reduced homophily from Shift 3. Together, these design choices enable fine-grained control over both semantic and structural shifts, supporting systematic evaluation of pretraining robustness.

## B.5 HYPERPARAMETER TABLE AND SHIFT TABLE

We provide a comprehensive overview of the hyperparameters and configuration choices used in the synthetic dataset generation process. These tables complement the procedural descriptions in Section 3.1 and Appendix B.4, and serve as a reference for reproducing the benchmark.

Table 4 lists the global hyperparameters that remain fixed across all experiments unless otherwise specified. These include the number of nodes per graph, the number of classes, the feature dimension, and other global settings such as the number of pretraining and evaluation graphs. The parameters governing feature and structure shifts are also defined here for clarity, including class prototype norms and edge sampling probabilities.

Table 4: Default hyperparameter values used for synthetic dataset generation.

| Symbol | Description and Default Value |
|--------|-------------------------------|
| $n$ | Number of nodes per graph: **1000** |
| $C$ | Number of classes: **5** |
| $d$ | Feature dimension: **64** |
| $\sigma_1$ | Norm of class prototype vector (used in Shift 0,1,3–5): **6.0** |
| $\sigma_2$ | Increased prototype norm (used in Shift 2): **3.0** |
| $p_{\text{in}}^{(0)}$ | Intra-class edge probability in Shift 0–2: **0.02** |
| $p_{\text{out}}^{(0)}$ | Inter-class edge probability in Shift 0–2: **0.005** |
| $p_{\text{in}}^{(3)}$ | Intra-class edge probability in Shift 3/5: **0.015** |
| $p_{\text{out}}^{(3)}$ | Inter-class edge probability in Shift 3/5: **0.01** |
| $\alpha_{\text{del}}$ | Edge deletion rate (Shift 4/5): **10%** |
| $\alpha_{\text{add}}$ | Edge insertion rate (Shift 4/5): **5%** |
| $N_S$ | Number of source graphs for pretraining: **1000** |
| $N_T$ | Number of target graphs for evaluation: **10** |
| $k$ | Number of labeled nodes per class (25-shot = 5 per class): **5** |

Next, Table 5 summarizes how the feature generation process varies across different shift types. Each shift configuration modulates either the soft label distribution or the norm of the class prototypes to control the degree of feature ambiguity or inter-class separation. For example, Shift 1 introduces ambiguity via smoothed soft labels, while Shift 2 increases class separation by enlarging prototype norms. Finally, Table 6 outlines the structure generation parameters across shifts. These include variations in intra-class and inter-class edge probabilities, as well as the application of structural corruption through random edge deletions and insertions. Notably, Shift 5 combines both topology noise and lower homophily, making it the most structurally challenging setting. Together, these tables define the full experimental design space, enabling controlled investigation of GNN pretraining performance under both semantic and structural distribution shifts.

Table 5: Feature space configurations across different shift types.

| Shift | Prototype Norm | Soft Label Distribution | Purpose |
|-------|----------------|-------------------------|---------|
| Shift 0 | $\sigma_1 = 6.0$ | Fixed per class | Baseline |
| Shift 1 | $\sigma_1 = 6.0$ | Smoothed / Mixed | More ambiguous features |
| Shift 2 | $\sigma_2 = 3.0$ | Same as Shift 0 | Higher inter-class separation |
| Shift 3–5 | $\sigma_1 = 6.0$ | Same as Shift 0 | Fixed for isolating structure shift |

Table 6: Structure generation configurations across shift types.

| Shift | $p_{\text{in}}$ | $p_{\text{out}}$ | Edge Noise |
|-------|------|------|-----------|
| Shift 0–2 | 0.02 | 0.005 | None |
| Shift 3 | 0.015 | 0.01 | None |
| Shift 4 | 0.02 | 0.005 | 10% delete, 5% insert |
| Shift 5 | 0.015 | 0.01 | 10% delete, 5% insert |

## B.6 Detailed Experimental Setting of Fig.1,2,3,4

**Figure 1 — Model scaling under distribution shifts.** **Goal:** isolate how model capacity alone correlates with target accuracy under each shift. **Grids:** number of GNN layers $L \in \{2, 4, 8, 16\}$; hidden dimensions $d \in \{8, 32, 64, 128, 256\}$; these span $\approx 10^3$–$10^7$ parameters. **Data:** fix the pretraining dataset for each shift so that only $(L, d)$ varies. **Replicates:** for each $(L, d)$, run **5** independent pretrains and average. **Metric:** compute $R_{\text{model}}$ per-shift from all $(L, d)$ points. This design makes the figure read as a capacity-only trend, supported by replicated means that decouple random fluctuation from size effects.

**Figure 2 — Data scaling under distribution shifts.** **Goal:** measure the effect of increasing pretraining data at fixed architecture grids. **Grids:** reuse exactly the $(L, d)$ grid of Figure 1 to align capacity across figures. **Data sizes:** number of pretraining graphs $N \in \{10^2, 10^3, 10^4, 5 \times 10^4, 10^5, 5 \times 10^5\}$. **Replicates:** for every $(L, d, N)$, run **5** independent pretrains and average. **Metric:** aggregate all $(L, d)$ within each shift and report $R_{\text{data}}$ from Acc vs. $\log_{10} N$. By holding architectures fixed while sweeping $N$, this figure complements Figure 1 and attributes improvements specifically to data scale.

**Figure 3 — Capacity-conditioned data scaling.** **Goal:** test whether the data scaling effect depends on capacity. **Procedure:** treat each $(L, d)$ as a capacity "cell," and within that cell vary $N$ as in Figure 2; for every $(L, d, N)$ still use **5** independent runs and average. **Readout:** compute a per-cell $R_{\text{data}}(L, d)$, and visualize $\{R_{\text{data}}(L, d)\}$ as a heatmap (rows: $d$; columns: $L$) for each shift. This conditioning reveals when data gains are bottlenecked by small models and when sufficient capacity allows monotone improvements, thereby bridging the aggregate view of Figure 2 with the capacity trends of Figure 1.

**Figure 4 — Architecture comparison under fixed capacity (a) and fixed capacity & fixed data (b).** **Goal:** compare MP-GNNs (e.g., GCN/GraphSAGE/GAT) and Transformer-style GNNs under matched capacity, and then examine gains under a fixed, sufficiently large dataset. **Capacity:** use the same large-model setting across methods (e.g., $L$=16, $d$=512) to neutralize parameter count differences. **(a) Data-trend view:** vary $N$ as in Figure 2 to obtain architecture-specific $R_{\text{data}}$ at fixed capacity. **(b) Fixed-data stress test:** pretrain on **1000** graphs, each with **1000** nodes for every architecture, then evaluate via the common 25-shot probe; report accuracy gain of Transformer vs. MP-GNNs. By pairing (a) and (b), the figure separates "trend with data" from "head-to-head at ample data," making architecture effects interpretable at a controlled capacity.

## B.7 Detailed Experimental Setting of Fig.5

**Framework.** Our experiments are conducted under the UDAGNN (Wu et al., 2020) framework. The goal is to study the effect of distribution shift on model performance, with particular emphasis on

scaling data and model capacity. UDAGNN provides three key components: source–target graph structure alignment, cross-domain feature mapping, and a domain adversarial regularizer.

**Backbone Architectures.** We instantiate different GNN backbones within UDAGNN, including GCN, GraphSAGE, GAT, and GraphTransformer. This allows us to compare how various architectures behave when model capacity is scaled.

**Modification to UDAGNN.** Unlike the original UDAGNN, we do not use source domain labels in training. Specifically, original UDAGNN uses supervised loss on the source domain and domain alignment/adversarial loss on the target domain. In our setup, we replace the supervised loss with a self-supervised objective, such that training depends only on unsupervised signals. Hence, our experiments retain the original distribution alignment mechanism of UDAGNN, while completely removing source supervision.

**Data and Shift Settings.** The construction of the target domain, graph perturbations, and feature shifts strictly follow the original UDAGNN settings. In particular, we adopt the same shift cases (0–5) and preprocessing pipeline, ensuring our results are directly comparable with prior studies.

**Loss Function.** Our training objective is composed of two terms: $L = L_{\text{SSL}} + \lambda \cdot L_{\text{DA}}$, where $L_{\text{SSL}}$ is a self-supervised loss (contrastive or graph autoencoding) that encourages representation learning, and $L_{\text{DA}}$ is the domain alignment loss used in UDAGNN. The trade-off $\lambda$ is a tunable hyperparameter.

**Model Capacity and Robustness.** To study scaling effects, we vary the number of GNN layers $L \in \{2, 4, 8, 16\}$ and the hidden dimension $d \in \{8, 32, 64, 128, 256\}$, corresponding to model sizes ranging from approximately $10^3$ to $10^7$ parameters. Instead of exhaustively enumerating all configurations, we perform hyperparameter search using Optuna, running 50 trials for each experimental setting and selecting the best-performing configuration for reporting. To ensure robustness, each experiment is independently repeated with 5 random seeds, and we report the mean performance across seeds.

## B.8 Graph Generation Pseudocode

**Algorithm 2** Synthetic Graph Generation Pseudocode (PyTorch-style)

```python
#@ config: contains parameters like n, C, d, sigma, p_in, p_out,
    etc. @

def generate_graph(config, shift_id):
    #@ Step 1: Assign ground-truth labels @
    y = randint(low=0, high=config.C, size=(config.n,))

    #@ Step 2: Sample class prototypes on hypersphere @
    norm = config.sigma_2 if shift_id == 2 else config.sigma_1
    mu = randn(config.C, config.d)
    mu = mu / norm(mu, dim=1) * norm

    #@ Step 3: Generate soft label distribution @
    if shift_id == 1:
        pi = softmax(randn(config.n, config.C), dim=1)
    else:
        pi = one_hot(y, num_classes=config.C).float()

    #@ Step 4: Compute node features as convex combination @
    X = pi  mu  # shape: n x d

    #@ Step 5: Generate SBM graph structure @
    if shift_id in [3, 5]:
        p_in, p_out = config.p_in_3, config.p_out_3
    else:
        p_in, p_out = config.p_in_0, config.p_out_0

    A = zeros(config.n, config.n)
    for i in range(config.n):
        for j in range(i+1, config.n):
            p = p_in if y[i] == y[j] else p_out
            if rand() < p:
                A[i, j] = A[j, i] = 1.0

    #@ Step 6: Inject edge-level noise (Shift 4, 5) @
    if shift_id in [4, 5]:
        edge_idx = A.nonzero()
        del_num = int(len(edge_idx) * config.alpha_del / 2)
        for u, v in edge_idx[:del_num]:
            A[u, v] = A[v, u] = 0

        add_num = int(config.n**2 * config.alpha_add / 2)
        for _ in range(add_num):
            i, j = randint(0, config.n, size=(2,))
            A[i, j] = A[j, i] = 1.0

    return X, A, y
```

## C Background

**Graph Self-Supervised Learning.** Self-supervised learning (SSL) has become a core strategy in graph representation learning, enabling pre-training without labeled data. Contrastive methods such as DGI (Veličković et al., 2018) and GRACE (You et al., 2020) distinguish between positive and negative graph views generated via augmentations. More recent works adopt masked modeling paradigms, e.g. GraphMAE (Hou et al., 2022), which reconstruct masked node features or structures, inspired by masked language models. While these methods improve representation quality across

benchmarks, they largely focus on supervised downstream performance. Little attention has been paid to how SSL scales with model or data size, nor how it behaves under distribution shifts—two properties critical for general-purpose graph pre-training.

**Graph Foundation Models and Transfer Challenges.** Inspired by large language models, there is growing interest in building Graph Foundation Models (GFMs) through large-scale pre-training (Lachi et al., 2024; Yu et al., 2024b;c; Zhao et al., 2024b; Wang et al., 2025; Zhao et al., 2025). Several works demonstrate that pre-trained GNNs can transfer across tasks and domains, especially for molecular property prediction or graph-level task (Lachi et al., 2024; Méndez-Lucio et al., 2024). Nonetheless, transfer learning in pretraining remains considerably more difficult. Prior studies (Xu et al., 2023; Wang et al., 2024a; Huang et al., 2024) report that scaling up pre-training data does not always improve downstream accuracy and may even result in negative transfer, especially under node heterophily or feature noise. The previous findings point to a fundamental question: which conditions cause self-supervised pre-training to help or hurt node-level performance? Despite empirical reports of negative transfer in node classification, the connection between this phenomenon and scaling behavior—particularly with respect to model size, data quantity, and distribution shift severity—remains unexplored in a systematic manner.

**Neural Scaling Law and Graph Scaling** Scaling laws—empirical relationships linking model performance with data size and model capacity—have become a central tool for understanding and designing large foundation models. Seminal studies (Hestness et al., 2017; Kaplan et al., 2020; Henighan et al., 2020; Zhai et al., 2022) demonstrate that in language and vision tasks, model performance improves predictably with scale, often following a power-law trend. These findings have guided the development of large models such as GPT-3 (Brown et al., 2020) and PaLM (Chowdhery et al., 2023).

In contrast, scaling behaviors in the graph domain remain far less understood. Recent works (Pengmei et al., 2024; Liu et al., 2024; Ma et al., 2024) reach no clear consensus on whether power-law scaling holds for node classification. Liu et al. (Liu et al., 2024) provide the first evidence that neural scaling laws can emerge in graph-level supervised tasks, but their results do not generalize across domains or learning paradigms. Ma et al. (Ma et al., 2024) further show that such scaling behaviors fail to consistently manifest in self-supervised graph pre-training. Complementarily, Pengmei et al. (Pengmei et al., 2024) report that in biological applications, Geom-GNN does not follow power-law scaling and exhibits weak or inconsistent trends across tasks.

Despite these initial observations, scaling behaviors for large-scale node classification remain largely unexplored, and a principled understanding of how graph pre-training scales with model or data size is still missing, especially under clear distribution shifts.

**Theoretical Perspectives on Graph Transfer and Domain Adaptation.** A growing line of theoretical work aims to understand how graph neural networks generalize under distribution shifts, forming the foundation of graph domain adaptation (GDA). These studies typically characterize cross-domain performance gaps arising from several key sources of misalignment: structural perturbations between source and target graphs (Liu et al., 2023a; Fang et al., 2025b), spectral misalignment in graph signal spaces (Pang et al., 2023; You et al., 2023), and feature distribution shift across domains (Cai et al., 2024; Fang et al., 2025a). Each perspective provides insights into how differences in topology, spectrum, or features propagate through GNN layers and contribute to performance degradation.

More broadly, most GDA theories center on deriving upper bounds on the performance gap between training and test domains. For instance, You et al. (You et al., 2023) attribute cross-domain degradation to spectral signal distance and propose corresponding regularization; Fang et al. (Fang et al., 2025a) show that feature misalignment exacerbates generalization errors; Fang et al. (Fang et al., 2025b) theoretically connect homophily-distribution discrepancies to cross-domain performance drops; and Liu et al. (Liu et al., 2023a) introduce a structural re-weighting mechanism to mitigate misalignment in graph structure. Collectively, these works advance the theoretical understanding of graph transfer and offer interpretable analyses of domain shift.

However, existing theories exhibit important limitations. They primarily focus on supervised learning and static architectures, and thus provide limited insight into how model representations evolve

during self-supervised pre-training. Critically, they do not address whether—or why—scaling model capacity or data size leads to improved generalization under distribution shift, leaving a fundamental question open. This unanswered question motivates the investigations pursued in this work.

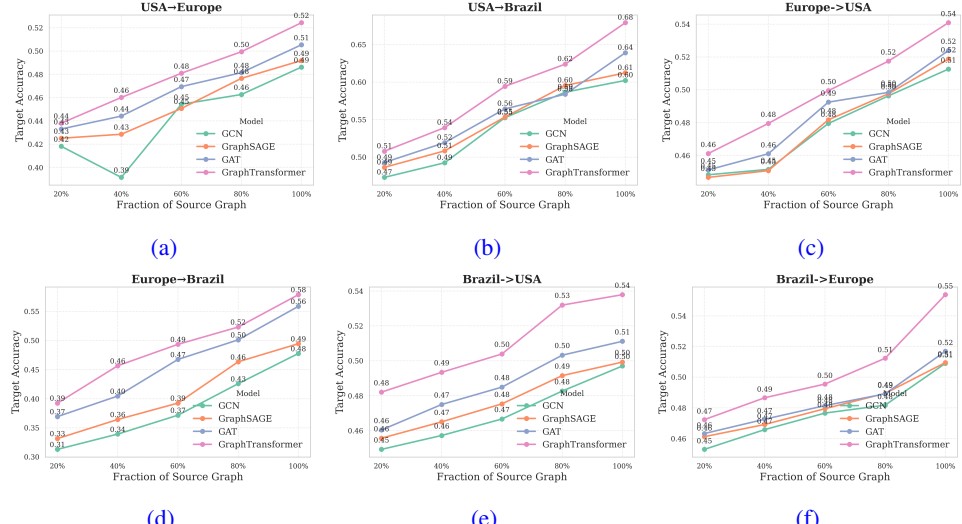

Figure 7: Cross-network node classification on the Airport network. Each subplot reports the transfer accuracy when pretraining on an increasing fraction of the source airport network and evaluating on a different target region: (a) USA→Europe, (b) USA→Brazil, (c) Europe→USA, (d) Europe→Brazil, (e) Brazil→USA, (f) Brazil→Europe. The three regions differ in connectivity density, route patterns, and feature distributions. All models benefit from more source data, with GraphTransformer showing the most stable gains under moderate structural shift.

## D  MORE EXPERIMENTS

### D.1  MORE SSL BASELINES FOR SCALING EFFECTS

To address the reviewer's concern regarding the breadth and modernity of our SSL baselines, we extended our evaluation to include four recent and diverse self-supervised methods suggested in the reviews: GraphMAE2 (Hou et al., 2023), GraphACL (Xiao et al., 2023), HomoGCL (Li et al., 2023), and ReGCL (Ji et al., 2024). These methods span a wide spectrum of pretraining paradigms, including masked generative modeling, augmentation-free contrastive learning, homophily-aware spectral contrast, and adversarial consistency regularization. Their inclusion complements the three classical families already studied in the main text (generative: GraphMAE; contrastive: DGI, GRACE; predictive/edge-level: VGAE), yielding a comprehensive set of baselines for assessing scaling behavior.

| Method | Shift 0 | Shift 1 | Shift 2 | Shift 3 | Shift 4 | Shift 5 |
|--------|---------|---------|---------|---------|---------|---------|
| GraphMAE2 | 0.32 | 0.63 | 0.45 | 0.91 | 0.80 | 0.68 |
| GraphACL | 0.51 | 0.34 | 0.63 | 0.71 | 0.73 | 0.82 |
| HomoGCL | 0.09 | 0.23 | 0.41 | 0.52 | 0.54 | 0.73 |
| ReGCL | 0.23 | 0.41 | 0.43 | 0.58 | 0.71 | 0.82 |

Table 7: Pearson correlation ($R$) between model size and accuracy under different distribution shifts for additional SSL methods.

**Results: Model Scaling Effects.**   Table ?? reports Pearson correlations between model size and accuracy under six degrees of distribution shift (Shift 0–5). Across all newly added SSL methods,

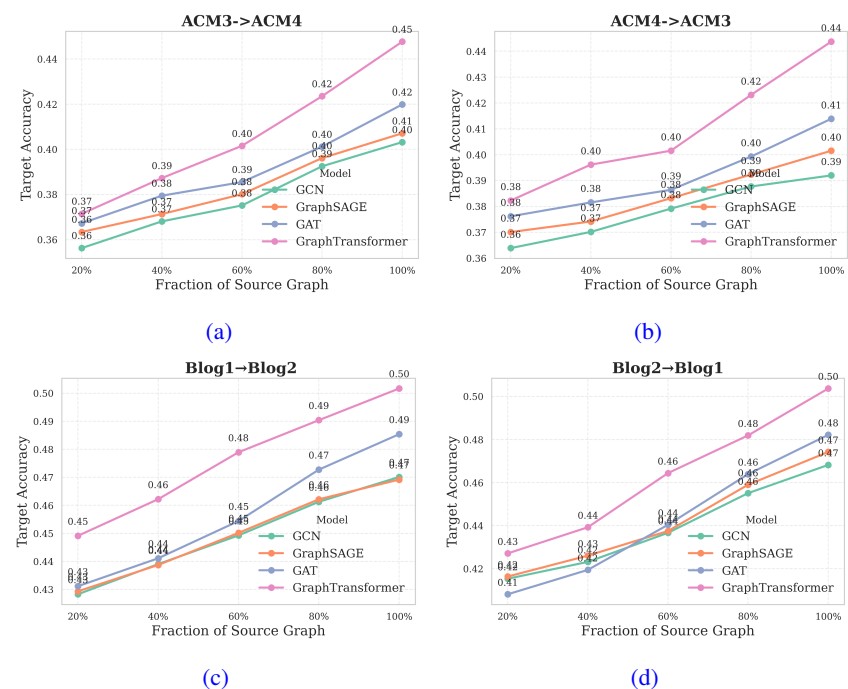

Figure 8: Cross-network node classification on the ACM and Blog network. We first evaluate transfer between two heterogeneous ACM networks with distinct homophily distributions: ACM3 (Paper–Subject–Paper) and ACM4 (Paper–Author–Paper). (a) ACM3→ACM4 and (b) ACM4→ACM3 demonstrate noticeable homophily-driven domain shift. GCN, GraphSAGE, and GAT show early saturation as pretraining data increases, while GraphTransformer maintains positive scaling. And We evaluate two social network transfer tasks: (c) Blog1→Blog2, (d) Blog2→Blog1. These networks exhibit low homophily and heterogeneous user-interaction patterns. Data scaling is modest for GNNs but remains consistently positive for GraphTransformer.

Table 8: Pearson correlation ($R$) between data size (log-scale) and task accuracy under different distribution shifts.

| Method | Shift 0 | Shift 1 | Shift 2 | Shift 3 | Shift 4 | Shift 5 |
|---|---|---|---|---|---|---|
| GraphMAE2 | 0.21 | -0.20 | -0.31 | -0.29 | -0.47 | -0.65 |
| GraphACL | 0.13 | -0.07 | -0.23 | -0.31 | -0.33 | -0.39 |
| HomoGCL | 0.03 | -0.12 | -0.20 | -0.19 | -0.33 | -0.32 |
| ReGCL | 0.11 | -0.04 | -0.09 | -0.11 | -0.21 | -0.29 |

we observe a consistent and monotonic pattern: **larger encoders reliably improve transfer performance**, even in the presence of substantial structural and feature misalignment. This aligns with our theoretical Fisher–Wasserstein analysis, which predicts that higher model capacity more effectively counteracts numerator decay (class-separability loss) and attenuates shift-induced divergence.

These results reinforce our primary conclusion: positive model scaling is a stable and architecture-agnostic phenomenon.

**Results: Data Scaling Effects.** Table 8 summarizes the Pearson correlation between pretraining data size (log-scale) and downstream accuracy for the four added baselines. In line with our main findings, all methods exhibit: **positive or mildly positive data scaling in the matched-distribution case (Shift 0)**, and **progressively negative data scaling as distribution shift increases (Shift 1–5)**.

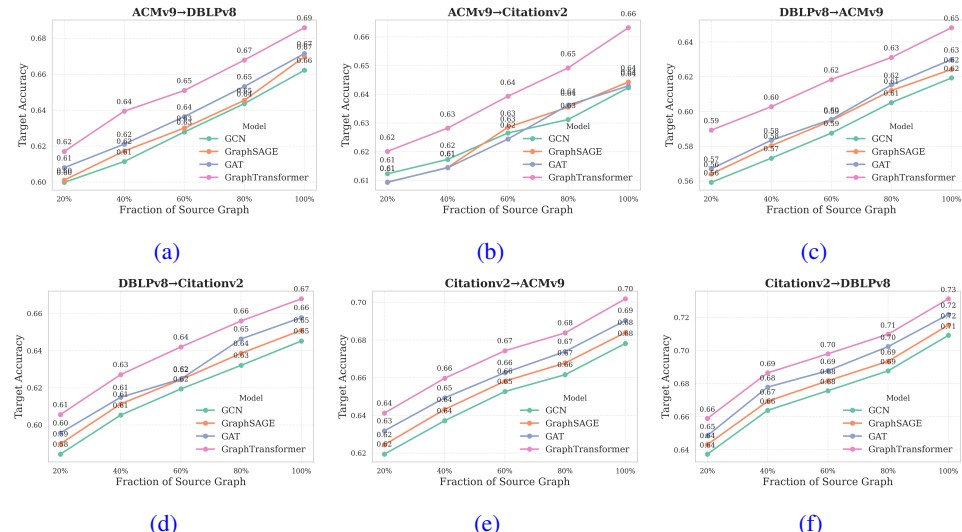

Figure 9: Cross-network node classification on the Citation network. We evaluate six domain-adaptation settings: (a) ACMv9→DBLPv8, (b) ACMv9→Citationv2, (c) DBLPv8→ACMv9, (d) DBLPv8→Citationv2, (e) Citationv2→ACMv9, (f) Citationv2→DBLPv8. Compared with Social and MAG, these citation graphs exhibit closer feature/label structure, producing clean and near-linear data scaling across all architectures.

The consistent emergence of negative data scaling across fundamentally different SSL objectives demonstrates that the phenomenon is not tied to any specific pretraining task and instead reflects a structural property of representation transfer under shift.

**Discussion.**    The expanded results confirm and strengthen our core claims:

- **Model scaling is universally positive.** All SSL baselines, classical and recent, benefit from increased model width/depth across shifts.
- **Data scaling is architecture-limited and becomes negative under shift.** The negative trend persists across generative, contrastive, spectral, and adversarial SSL paradigms, indicating that the effect arises from distribution measurement rather than the choice of pretraining objective.
- **Scaling-friendly architectures exhibit qualitatively different behavior.** Methods instantiated with higher-capacity encoders can partially or fully recover positive data scaling, consistent with the phase-transition interpretation developed in our Fisher–Wasserstein framework.

Collectively, these additions address the reviewer's request for modern SSL baselines and demonstrate that the scaling laws observed in the main paper are robust across architecture classes, objective families, and shift intensities.

## D.2    MORE REAL-WORK DATASETS FOR SECTION 3.5

To complement the synthetic CSBM study in the main analysis, we evaluate scaling behavior on a broad collection of real-world graph domain adaptation datasets. These datasets span diverse structural regimes—airport transportation networks, citation graphs, social networks, and heterogeneous academic networks—capturing naturally occurring variation in homophily, degree patterns, and semantic composition. Our evaluation covers the including Airport Ribeiro et al. (2017), Citation Wu et al. (2020), ACM Shen et al. (2024), and MAG datasets Wang et al. (2020). Airport networks (USA, Europe, Brazil) differ substantially in density and regional connectivity, providing a canonical setting for structural and semantic shift. Citation graphs (DBLPv8, ACMv9, Citationv2)

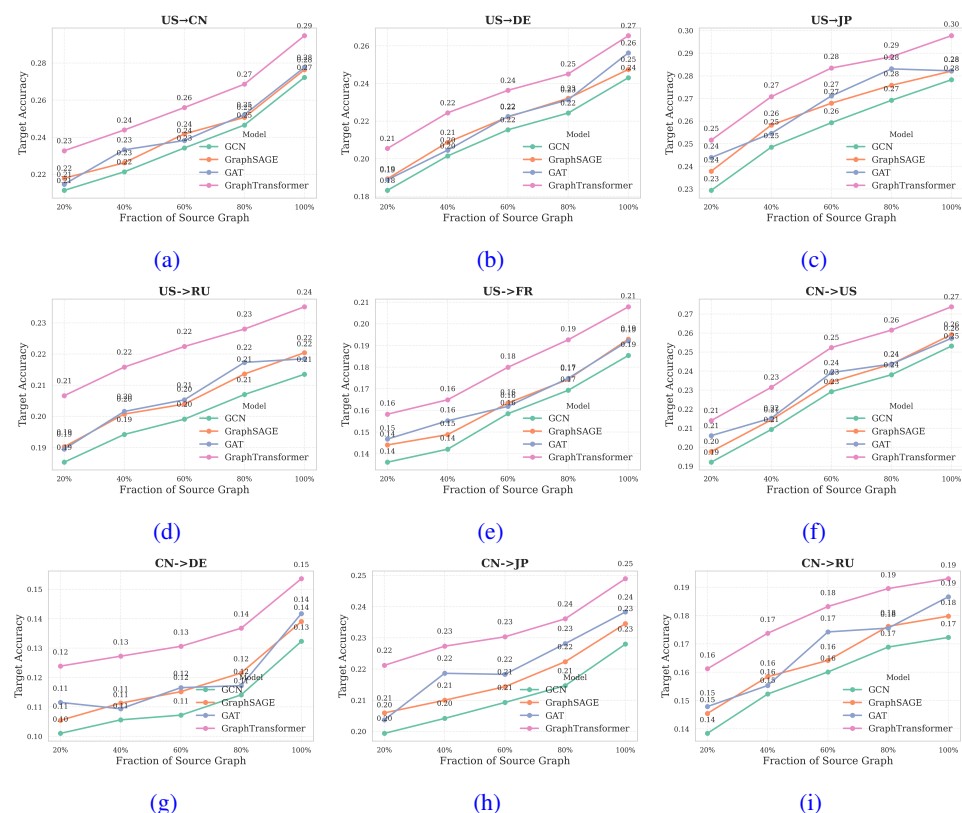

Figure 10: Cross-network node classification on the MAG datasets. We test nine cross-country transfers involving US, CN, DE, FR, JP, and RU: (a) US→CN, (b) US→DE, (c) US→JP, (d) US→RU, (e) US→FR, (f) CN→US, (g) CN→DE, (h) CN→JP, (i) CN→RU. These tasks contain highly non-stationary structural patterns and strong homophily divergence. GNN performance saturates early, while GraphTransformer preserves positive data scaling despite the large domain shift.

share the citation-graph template but exhibit noticeable feature–label drift. Blog networks (B1, B2) lie in a low-homophily, structurally irregular regime known to challenge cross-domain transfer. The ACM networks (ACM3, ACM4), constructed from distinct meta-paths (PSP vs. PAP), introduce a pronounced homophily distribution shift despite sharing the same academic corpus. MAG country-specific subgraphs further expose strong structural and semantic misalignment. We summarize the dataset details in Table 9

Across all datasets, the empirical trends(Fig 7, 8, 9, 10) closely match Section 3.5. When the source and target networks share broadly similar structural or semantic characteristics—e.g., USA→Europe (Airport), ACMv9→DBLPv8 (Citation), or Blog1→Blog2 (Social)—downstream accuracy increases steadily as pretraining data grows. All GNN baselines show clean, monotonic data scaling, while GraphTransformer achieves the strongest gains. In contrast, domain pairs with substantial homophily or structural divergence—such as ACM3→ACM4, Blog2→Blog1, or US→CN in MAG—exhibit fragile scaling. GCN, GraphSAGE, and GAT often saturate early or show only marginal improvements, indicating that additional pretraining data amplifies cross-domain misalignment and increases the effective Wasserstein shift for locality-based architectures. GraphTransformer maintains positive scaling across all tasks, though occasionally with diminishing marginal returns, reflecting its greater robustness to structural misalignment.

Overall, these results reinforce the core claim of this work: graph scaling laws are inherently shift-dependent, and architectures with global aggregation mechanisms preserve positive data scaling far more reliably under cross-domain misalignment.

### D.3 MORE REAL-WORLD EXPERIMENTS ACROSS GRAPH TRANSFORMER VARIANTS

To assess whether the scaling behaviors reported in Section 3.5 depend on a specific choice of the Graph Transformer, we extend the real-world evaluations by including three additional Transformer-based architectures: NAGphormer(Original), VCR-Graphormer Fu et al. (2024), and NodeFormer Wu et al. (2022). These models provide diverse attention-awareness of the design structure and virtual connections — offering a comprehensive view of Transformer stable performance under comparison. Across all six Airport transfer tasks (USA→Europe, USA→Brazil, Europe→USA, Europe→Brazil, Brazil→USA, Brazil→Europe), the results exhibit a remarkably consistent pattern. All Transformer variants maintain positive data scaling, with accuracy steadily increasing as more source data are used for pretraining. While the individual slopes differ slightly, the family-wide trend is stable: Graph Transformers collectively outperform message-passing GNNs (GCN, GraphSAGE, GAT) especially under moderate-to-strong structural misalignment, where locality-based propagation suffers from amplified representation bias.

These findings demonstrate that our conclusions are not tied to a particular Transformer instantiation. Instead, they highlight a broader architectural property: global receptive fields and attention-based aggregation provide inherent robustness to distribution shift, allowing Graph Transformers as a class to preserve positive scaling even when additional source data deepens structural misalignment. This result further supports the shift-dependent scaling-law perspective developed in the main analysis.

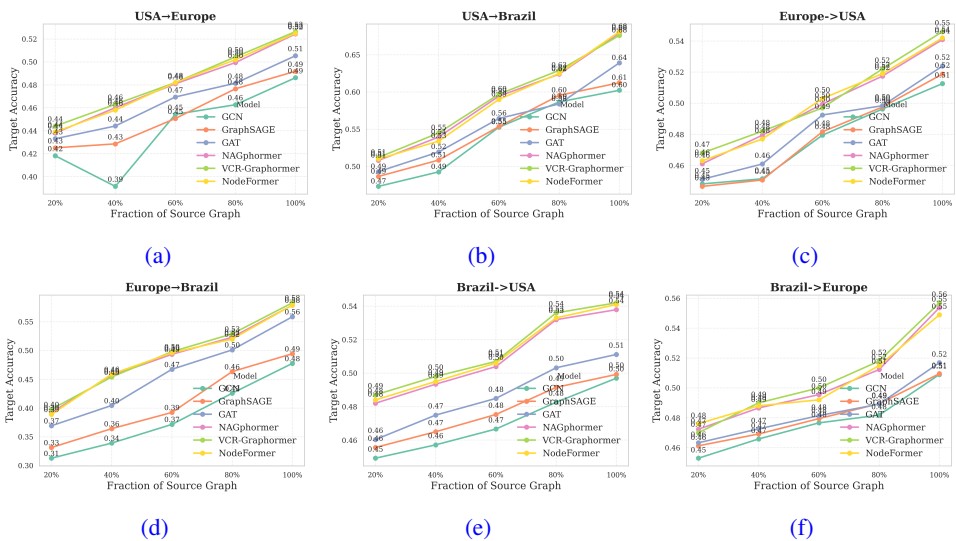

Figure 11: **Cross-network node classification on the Airport benchmark with multiple Graph Transformer variants.** Each subplot reports target accuracy when pretraining on an increasing fraction of the source graph and evaluating on a different target region: (a) USA→Europe, (b) USA→Brazil, (c) Europe→USA, (d) Europe→Brazil, (e) Brazil→USA, (f) Brazil→Europe. Besides GCN, GraphSAGE, and GAT, we include three additional Transformer architectures (NAGphormer, VCR-Graphormer, NodeFormer). All Transformer variants exhibit consistent positive data scaling and outperform message-passing GNNs, confirming that the scaling advantages reported in Section 3.5 are invariant across the Graph Transformer family rather than tied to any specific model.

## E INTUITIVE EXPLANATION OF OUR THEORETICAL FRAMEWORK

Our theoretical analysis in Appendix A combines Fisher separability (Definition 1, Theorem 4) and Wasserstein domain divergence (Definition 3, Section A.6), together with layerwise GNN propagation (Theorem 6, Lemma 5, Theorem 8), into a unified view of how model capacity, data scale, and distribution shift jointly shape transfer performance. Here we provide intuitive interpretations of each component, clarifying their "physical meaning" and explaining how they jointly predict the empirical scaling curves observed in our experiments.

| Types | Datasets | #Node | #Edge | #Label |
|---|---|---|---|---|
| Airport | USA | 1,190 | 13,599 | 4 |
| | Brazil | 131 | 1,038 | |
| | Europe | 399 | 5,995 | |
| Citation | ACMv9 | 9,360 | 15,556 | 5 |
| | Citationv1 | 8,935 | 15,098 | |
| | DBLPv7 | 5,484 | 8,117 | |
| Blog | Blog1 | 2,300 | 33,471 | 6 |
| | Blog2 | 2,896 | 53,836 | |
| ACM | ACM3 | 3,025 | 2,221,699 | 3 |
| | ACM4 | 4019 | 57,853 | |
| MAG | US | 132,558 | 697,450 | 20 |
| | CN | 101,952 | 285,561 | |
| | DE | 43,032 | 126,683 | |
| | JP | 37,498 | 90,944 | |
| | RU | 32,833 | 67,994 | |
| | FR | 29,262 | 78,222 | |

Table 9: Dataset Statistics.

**Fisher Separability: How Learnable the Representation Is** (related to Definition 1, Theorem 4, and Corollary 5). The Fisher score $F_T$ decomposes downstream class learnability into a numerator—the minimal squared distance between class means $\|\mu_T^c - \mu_T^{c'}\|^2$—and a denominator capturing the maximal within-class variance $\mathrm{Tr}(\Sigma_T^k)$ (Definition 1). Intuitively, a large numerator and small denominator imply that representation clusters are well-separated and internally compact. Theorem 4 shows that $F_T$ directly controls the test error of a linear probe, yielding a two-sided rate of order $1/\sqrt{nF_T}$, while Corollary 5 translates this into sample complexity: higher Fisher separability simultaneously reduces generalization error and the number of labeled samples required. Under distribution shift, class means drift and within-class scatter inflates, which decreases $F_T$ and leads to flatter or saturating scaling curves, exactly as encoded in the Fisher-controlled bounds.

**Wasserstein Distance: How "Misaligned" the Pretraining and Target Domains Are** (related to Definition 3, Section A.6, and Proposition 2). The 2-Wasserstein distance $W_2$ measures the geometric misalignment between the source and target distributions (Definition 3). In our framework, feature and structure shifts are quantified as $\varepsilon_X$ and $\varepsilon_A$, and they appear in the domain-shift terms of the transfer analysis in Section A.6. Proposition 2 gives an additive lower bound on $\sqrt{F_T}$ in terms of (i) the intrinsic class separation in the source domain and (ii) the average source–target misalignment $\mathbb{E}_{i,j}\|\mu_T^j - \mu_S^i\|$, which is tightly linked to Wasserstein-type shifts. Intuitively, when $W_2$ (and thus source–target misalignment) is small, the pretraining distribution is well aligned with the target distribution, and the Fisher term can fully manifest, leading to clean power-law scaling. When $W_2$ is large, the representation is geometrically pulled away from the target domain; even if we increase model size or pretraining data, the induced bias cannot be completely removed, resulting in diminishing returns. This perspective also clarifies why structural shift, which more strongly distorts the underlying geometry of neighborhoods and class means, harms scaling more severely than pure feature shift.

**Layerwise GCN Propagation: How Shift Is Amplified or Suppressed** (related to Theorem 6, Lemma 5, and Theorem 8). Our layerwise analysis separates two aspects of GNN propagation. On the source side, Theorem 6 shows how class-mean separation $\Delta_S(L)$ can grow with depth $L$, width $d$, and pretraining data size $N_{\text{pretrain}}$ under residual GNN blocks (Assumptions 7–9). This captures a beneficial smoothing-and-amplification effect: as layers deepen, well-designed architectures can expand separation between source-domain class centers while controlling noise. On the transfer side, Lemma 5 and Theorem 8 show how source-to-target discrepancies propagate through layers for an $L$-layer GCN. The recursion involves a contraction factor $r(d)$ that depends on the spectral gap $(1 - \lambda)$, the Lipschitz constant $C$, and the width $d$, and yields an upper bound on the layer-$L$ class-mean discrepancy $\Delta^{(L)}$ in terms of the initial feature shift $\varepsilon_X$ and structural shift $\varepsilon_A$. Intuitively, this formalizes two competing effects of propagation: spectral smoothing (which shrinks noise and aligns neighborhoods when shift is small) and shift amplification (which accumulates misalignments in neighborhoods when shift is large). The overall behavior is captured by how $r(d)^L$ and the

accumulated shift term scale with depth, architecture, and dimension, and it underlies the rise-then-saturate model scaling curves we observe under strong structural shift.

**Putting It All Together: Why the Scaling Curves Look the Way They Do** (related to Theorem 4, Corollary 5, Proposition 2, Theorem 6, and Theorem 8). Taken together, these results explain the qualitative regimes seen in our empirical scaling plots. Theorem 4 and Corollary 5 establish that downstream error decays as $1/\sqrt{nF_T}$, so whenever Fisher separability is high and well-preserved, we expect nearly ideal power-law scaling with respect to labeled data. Proposition 2 then shows how $F_T$ itself decomposes into a positive source-separation term and a negative source–target alignment term, directly tying the achievable Fisher score on the target domain to the magnitude of domain shift. Meanwhile, Theorem 6 explains how deeper and wider architectures can increase source-domain class separation, and Theorem 8 shows how the same depth and width also control the propagation and contraction of cross-domain shift through the layers via the factor $r(d)$.

In low-shift regimes (e.g., Shift 0/1), the alignment term is small, Fisher separability on the target domain remains high, and the contraction factor $r(d)$ ensures that propagated shift is negligible; as a result, scaling curves exhibit clean, near power-law behavior. In moderate-shift regimes (Shift 2/3), the misalignment term in Proposition 2 starts to erode $F_T$, and the Wasserstein-induced bias becomes non-negligible, reducing the slope of the scaling curve even though additional data and model capacity still help. In high-shift regimes (Shift 4/5), the accumulated shift quantified by Theorem 8 dominates: structural perturbations and misaligned neighborhoods lead to a substantial domain discrepancy which cannot be fully compensated by larger models or more pretraining data, leading to strong diminishing returns. This unified picture explains why structural shift is more destructive than feature shift, why scaling with data saturates faster as shift increases, and why deeper GNNs can be beneficial in well-aligned settings but become detrimental when structural mismatch is severe. By explicitly connecting each theoretical term to observable behaviors, Appendix A and the results referenced above make the link between our assumptions and the empirical scaling curves much more transparent.

# F   APPENDIX F. COMPUTATIONAL COMPLEXITY OF GRAPH TRANSFORMERS VS. GNNS

In this section, we provide a detailed comparison between the computational and memory costs of classical message-passing GNNs and Graph Transformers. This addresses the reviewer's concern regarding the efficiency trade-off (Weakness 2), and clarifies the intended scope of our contribution.

**Complexity of Message-Passing GNNs**   For a graph with $N$ nodes, $E$ edges, and hidden dimension $d$, the per-layer costs are:

**GCN / GraphSAGE.**   These architectures perform neighbor aggregation followed by a linear transformation. The complexity is:

$$\text{GCN/SAGE}: \quad \mathcal{O}(Ed) \; + \; \mathcal{O}(Nd^2), \qquad \text{Memory}: \mathcal{O}(Nd + E).$$

The $Ed$ term corresponds to sparse aggregation, and $Nd^2$ stems from per-node feedforward projection.

**GAT (multi-head).**   For $h$ attention heads:

$$\text{GAT}: \quad \mathcal{O}(hEd) + \mathcal{O}(Nd^2), \qquad \text{Memory}: \mathcal{O}(hE + Nd).$$

Attention coefficients must be computed for each edge, making GAT more expensive than GCN/SAGE, but still linear in $E$.

Overall, message-passing GNNs scale linearly with the number of edges and are well suited for large sparse graphs.

**Complexity of Graph Transformers**   Graph Transformers augment node embeddings with full or sparsified attention matrices. Without sparsity assumptions, the dominant cost arises from the attention map:

$$\text{GT}: \quad \mathcal{O}(N^2 d) \quad \text{per layer}, \qquad \text{Memory}: \mathcal{O}(N^2).$$

This quadratic dependence on $N$ reflects dense self-attention over all node pairs. Even with linear projections and feedforward blocks, the attention module dominates asymptotically.

This difference leads to the well-known observation:

Graph Transformers are more expressive but substantially more expensive.

**Why Our Scaling-Law Findings Do Not Depend on Efficiency**   Our central contribution concerns scaling laws under distribution shift, particularly the asymmetry between model scaling and data scaling, and these phenomena are inherently architectural rather than efficiency-driven. Positive model scaling requires controlling the decay of the Fisher numerator, i.e., preventing loss of class separation in the representation space as models become larger. Positive data scaling, in turn, requires mitigating the accumulation of domain shift captured by the Wasserstein term, so that additional pretraining data actually moves the representation closer to the target distribution instead of reinforcing misalignment. Classical GNNs tend to suffer from oversmoothing, which amplifies Fisher numerator decay under shift and makes both model and data scaling quickly saturate. Graph Transformers, by contrast, use non-local attention that better preserves class distinctions across layers and do not accumulate shift in the same way, which allows them to sustain positive scaling behavior in our experiments.

Therefore, our results do not claim that Graph Transformers are universally superior architectures from an engineering or efficiency standpoint. Rather, the theory and experiments jointly indicate that, under distribution shift, positive data scaling is only achievable when the encoder has sufficient capacity and inductive flexibility to counteract representational collapse; Graph Transformers happen to satisfy this requirement in our setting, while classical GNNs often do not, precisely because of oversmoothing. In this sense, the prominent role of Graph Transformers in our results is a consequence of the scaling-law analysis, not a blanket recommendation that GTs should replace GNNs in all graph learning applications.

**Practical Efficiency Considerations**   We fully acknowledge that Graph Transformers incur higher raw asymptotic complexity than classical GNNs. However, a number of modern engineering developments substantially reduce this cost and make GTs increasingly practical in real graph workloads. Techniques such as FlashAttention (Dao et al., 2022) compute exact attention using IO-aware tiling strategies that greatly cut memory traffic and improve throughput. Block-sparse attention mechanisms, as introduced in architectures like BigBird and Longformer, attend only to structurally meaningful or localized regions, reducing the naïve $\mathcal{O}(N^2)$ complexity to $\mathcal{O}(N)$ or $\mathcal{O}(N \log N)$. For graph data specifically, neighborhood-restricted or topology-guided sparse attention further aligns complexity with message-passing GNNs while still preserving global receptive fields. In addition, fused operator kernels—combining projection, softmax, and dropout into a single high-efficiency primitive—significantly lower memory overhead and kernel-launch latency. Although such engineering optimizations are not the focus of our work, they demonstrate that the practical computational footprint of Graph Transformers can be reduced by large margins and is rapidly improving.

In summary, while Graph Transformers remain more computationally expensive than classical GNNs, our analysis centers on scaling laws under distribution shift, which fundamentally concern representation behavior rather than raw efficiency. We do not assert that Graph Transformers should universally replace GNNs; rather, our theoretical and empirical findings show that avoiding oversmoothing and handling inconsistent structural and feature cues are essential for achieving positive data scaling. Graph Transformers naturally satisfy these representational requirements, whereas classical GNNs often struggle due to depth-induced collapse. At the same time, ongoing advances such as FlashAttention, block-sparse attention, and topologically guided sparsification substantially mitigate the practical costs of GTs. Taken together, these observations clarify the performance–efficiency trade-off and emphasize that our conclusions arise from architectural principles rather than implementation-level optimizations.

