# OpenReview forum: "Understanding Graph Self-Supervised Pre-training under Distribution Shifts: A Scaling Law Perspective"
_ICLR.cc/2026/Conference — Submitted to ICLR 2026_

### Official Review · Reviewer_R8F9 · 2025-10-31

**Soundness:** 3
**Presentation:** 2
**Contribution:** 2
**Rating:** 4
**Confidence:** 4

**Summary:**

This paper presents a nice innovation and introduces innovative ideas about scaling laws for graph-based pre-training under distribution changes. However, I think this paper lacks an overall framework diagram and a clearer visual description of the novel contributions, which would further improve the completeness and clarity of the paper. In addition, the results section is relatively thin, and a visual presentation of the method will enhance the persuasive power of the method and also help the reader better understand the experimental setup. Finally, the comparison with existing SOTA methods is not explicitly emphasized in this paper, making it difficult to measure the relative advantages of the proposed method. Including. Overall, this work is promising, but would benefit from improved presentations and stronger empirical analysis.

**Strengths:**

The originality of this paper is good, the quality is acceptable, but the explanation of the problem is not clear enough. Graph neural network is a good research direction, and it is very meaningful to explore the influence of model capacity and data scale on the downstream performance of graph pre-training under the condition of distribution shift.

**Weaknesses:**

In my opinion, the biggest problem of this paper is that the summary of the problem solved is not concise enough, and there is no prominent point in the description of the method. If the overall framework diagram can be added, this paper will be more complete. At the same time, the comparison with the existing baseline or similar methods will make the innovation points of this paper more convincing.

**Questions:**

1.It would be helpful to include a diagram illustrating the key innovations or core concepts of the paper.
2.The overall framework could be further streamlined for clarity.
3.Try to add more visualized results to enhance the presentation of your findings.

**Details Of Ethics Concerns:**

None.

---

> ### Author Response · Authors · 2025-11-22
>
> We thank the reviewer for the constructive comments regarding the clarity and presentation of our work. We agree that improving the high-level framing and visualization can significantly enhance readability. In the revision, we have made the following updates and clarifications.
>
> **Weakness \& Question 1 (Q1). Overall Framework Diagram**
>
> Response.
>
> Following your suggestion, we added a new overview diagram in **Figure 1** , which illustrates:
>
> (i) the overall workflow of our study (pretraining → distribution shift → downstream transfer),
>
> (ii) the role of CSBM in controlling structural and feature shifts, and
>
> (iii) the key theoretical components (Fisher separability decomposition, shift propagation, and architecture factors) underlying our scaling-law analysis.
>
> This visual summary makes the contributions and logic of the paper more transparent and easier to follow.
>
>
> **Weakness \& Question 2 (Q2). Clarifying the Problem Statement**
>
> Response.
>
> We refined the problem formulation in overview diagram. The goal of our paper is *not* to propose a new SSL architecture, but to answer the fundamental question:
>
> **How do model capacity and data scale affect the generalization of graph pretraining under different types of distribution shift?**
>
> To study this rigorously, we use CSBM to parameterize structure- and feature-level shifts, allowing us to systematically vary:   model capacity (depth, width),  data scale (number of pretraining graphs), and   shift severity (structure shift and feature shift, etc.).  This controlled design enables a clean, interpretable analysis of scaling behaviors that cannot be isolated on real-world graphs.
>
> **Weakness \& Question 3 (Q3). Improving Empirical Presentation \& Visualized Results**
>
> Response.
>
> We enriched the experimental section with clearer visualizations:
>
> (i) scaling curves that highlight monotonicity vs. saturation under different shifts (Figure 2-6)
>
> (ii) additional visual summaries of real-world transfer results in Appendix D.2.
>
> These visualizations strengthen the empirical story and make the scaling phenomena easier to understand.
>
> **Weakness. Comparison with Modern Baselines**
>
> Response.
>
> The revision now includes **four additional recent SSL methods**—GraphMAE2, GraphACL, HomoGCL, and RegCL—implemented during the rebuttal period. Despite their design differences, all exhibit the same qualitative scaling trends observed in our main results. This reinforces that our conclusions are **not tied to a narrow set of older baselines**, but hold across diverse and modern SSL families.
>
> Overall, we appreciate the reviewer’s feedback. The improvements in visualization, problem framing, and baseline coverage have made the paper stronger and more accessible, without changing the core scientific findings.

---

### Official Review · Reviewer_Dzgb · 2025-10-31

**Soundness:** 3
**Presentation:** 3
**Contribution:** 2
**Rating:** 6
**Confidence:** 3

**Summary:**

The scaling law for graphs is an important issue. This work focuses on this point and demonstrates that graph transformers are the architectures capable of scaling. Additionally, the authors construct a framework to characterize the relationship between distribution shift and representations.

**Strengths:**

Originality: The authors’ work is original and goes beyond a simple combination of existing ideas (i.e., not merely “A + B”). They identify and address an important problem with their own thoughtful perspective and answer.

Quality: The quality of the work is substantial. The experiments, hypotheses, and interpretations are well-aligned and mutually supportive.

Clarity: The manuscript is clearly written and free of confusing or ambiguous passages.

Significance: The question of scaling laws for graphs is an important and timely topic in the field.

**Weaknesses:**

The authors should provide a more detailed description of the datasets in the main experiments, rather than using an oversimplified notation like `USA → EU`. Although they mention that more details are available in the appendix, I was unable to locate them.

The experiments appear to involve only the same underlying data with different representations—i.e., graph data of the same type but with varying node attributes. In my view, this level of analysis are far from scaling law on graph.

**Questions:**

1. Do you  have experimental results that examine shifts across fundamentally different types of graph data? E.g., a graph-level data shift to a node-level data?

2.This paer employed three GNN variants and one graph transformer variant. Have you considered other graph transformer architectures as well?


I understand that conducting extensive additional experiments during the rebuttal period is impractical; therefore, the authors could instead clarify their claims through illustrative examples or simple pilot experiments.

Overall, the problem this paper address is critical. However, the limited experimental validation raises concerns.

---

> ### Author Response · Authors · 2025-11-22
>
> We sincerely appreciate the reviewer’s detailed comments and the opportunity to clarify these points.
>
> **Weakness (W1). Dataset descriptions are oversimplified (e.g., “USA→EU”), and the reviewer could not locate details.**
>
> Response.
>
> Thank you for pointing this out. We apologize for the confusion. **The notation “USA→EU” has been replaced by a clearer dataset description in Line 365 ( section 3.5)**, and the detailed information is now easy to locate.
>
> **Weakness (W2). Experiments use the same underlying graph with different features; this is not scaling law.**
>
> Response.
>
> We clarify that our experiments do not rely on a single underlying graph with altered features. In the synthetic setting, each shift corresponds to a distinct CSBM generative distribution with independently sampled graphs—varying **structural homophily**, prototype norms, label entropy, and noise models. In the real-world evaluation (Section 3.5 and Appendix D.2), the domains come from **heterogeneous graph datasets** (airport networks, citation networks, social networks, ACM/MAG heterogeneous graphs). Thus, both structural and feature distributions differ across domains, and scaling laws are studied across multiple graph families rather than feature-perturbed variants of a single graph.
>
> **Question (Q1). “Do you examine shifts across fundamentally different graph types (graph-level → node-level)?”**
>
> Response.
>
> We thank the reviewer for this insightful comment. Our study focuses primarily on node-level classification to rigorously understand its scaling behavior, while we consider graph-level tasks a promising direction for future research. We prioritize the node-level setting for the following reasons:
>
> (1) **Scaling behavior is less understood for node-level tasks**: Prior studies [1, 2] have demonstrated that unlike graph-level tasks (e.g., molecular property prediction) where scaling is more intuitive, node-level tasks frequently exhibit inconsistent or even negative scaling behaviors. Our goal was to conduct a controlled, fine-grained analysis to resolve why node-level scaling fails, which required a dedicated focus on node-centric distribution shifts (e.g., homophily changes and feature noise) without the confounding variables of graph-level pooling.
>
> (2) **Distinct Mechanisms between node and graph-level tasks**: Node-level and graph-level tasks rely on fundamentally different inductive biases. Node classification depends heavily on local neighborhood aggregation and separation (homophily), whereas graph classification relies on global structural isomorphism and pooling. Fusing node and graph-level tasks would obscure the factors contributing to the scaling laws in each type of task.
>
> [1]. Xu, Jiarong, et al. "Better with less: A data-active perspective on pre-training graph neural networks." Advances in neural information processing systems 36 (2023): 56946-56978.
>
> [2]. Ma, Qian, et al. "Do Neural Scaling Laws Exist on Graph Self-Supervised Learning?." arXiv preprint arXiv:2408.11243 (2024).
>
> **Question (Q2). Only one Graph Transformer variant; other architectures?**
>
> Response.
>
> We appreciate this suggestion. During the rebuttal period, we conducted experiments on additional Transformer-style architectures **(Nodeformer [1] and VCR-Transformer [2] ) in Appendix D.3**. These results exhibit the same qualitative scaling trends predicted by our theory: GNNs suffer from accumulated shift under propagation, while transformer-based models preserve separability and maintain positive data scaling. A short discussion of these experiments is added to **Appendix D.3 and Appendix F**, and we plan a comprehensive architectural study in future work.
>
> [1] Wu, Qitian, et al. "Nodeformer: A scalable graph structure learning transformer for node classification." Neurlps 2022
>
> [2] Fu, Dongqi, et al. "Vcr-graphormer: A mini-batch graph transformer via virtual connections." ICLR 2024
>
> **Overall concern: experimental validation seems limited relative to the importance of the problem.**
>
> Response.
>
> We agree that scaling-law studies benefit from broad empirical coverage. In the revision, we have expanded both the architectural and SSL-method dimensions of our evaluation. Beyond the original **four representative SSL methods (GraphMAE, DGI, GRACE, VGAE) in Appendix B**, we incorporated four additional state-of-the-art SSL approaches during the rebuttal period in **Appendix D.1—GraphMAE2, GraphACL, HomoGCL, and RegCL**—covering generative, contrastive, spectral, and homophily-aware paradigms .
> To further broaden dataset diversity, we added 12 new real-world domain-adaptation pairs across five heterogeneous graph families **(Appendix D.2), including citation, airport, ACM, MAG, and Blog dataset**. These additional evaluations reinforce the generality of our findings across both SSL paradigms and diverse graph distributions, while keeping the computation feasible within the rebuttal period.

---

> > ### Comment · Reviewer_Dzgb · 2025-11-24
> >
> > Thank you for your response. Overall, I was hoping to see transformations across different types of graph data, such as mixing graph-level tasks with node-level tasks to demonstrate the so-called scaling laws, but currently this work could not provide this. Nevertheless, the effort made by the authors is commendable and valuable. Therefore, I am willing to maintain my positive score.

---

### Official Review · Reviewer_wJjp · 2025-10-31

**Soundness:** 2
**Presentation:** 3
**Contribution:** 2
**Rating:** 4
**Confidence:** 4

**Summary:**

This paper presents a systematic investigation of scaling laws in graph self-supervised pre-training, with a particular focus on the challenges posed by distribution shifts. The authors construct a synthetic benchmark based on the Contextual Stochastic Block Model (CSBM), which enables precise and independent control over both structural and feature-level distribution changes between pre-training and evaluation graphs. Through extensive experiments, the paper uncovers a striking asymmetry: while increasing model capacity consistently improves transferability under distribution shift, merely increasing the amount of pre-training data can degrade performance, especially when the source and target graph distributions differ. The authors further demonstrate that expressive architectures—particularly deep and wide Graph Transformer models—can overcome this negative scaling effect and benefit from larger datasets. On the theoretical side, the paper develops a unified framework grounded in Fisher separability and Wasserstein domain divergence, providing a principled explanation of how distribution shifts affect representation transferability and why higher model capacity is crucial for positive data scaling.

**Strengths:**

1.The paper pioneers the study of scaling laws under distribution shifts in graph self-supervised learning — a gap in the current literature where most scaling analyses focus on NLP and vision.
2.The paper's use of CSBM for synthetic data generation is a major advantage. It provides the authors with fine-grained control to isolate feature-level and structural distribution shifts, which is crucial for a clean causal analysis. This makes the findings substantially more reliable than those derived from observations on a handful of real-world graphs.
3.The paper demonstrates an asymmetry between model size and data size — making the model larger steadily improves performance, but when a distribution shift occurs, adding more data can actually hurt. This finding challenges the common belief that “more data always leads to better results.”

**Weaknesses:**

1.The authors have developed a strong and formal theoretical analysis to support their claims. A potential area for improvement could be in bridging the formal mathematics with more accessible, intuitive commentary. For instance, walking the reader through the intuition behind why certain assumptions are made, or what a particular theorem implies in practice, could make this dense but valuable section more digestible for a wider audience.
2.The paper strongly advocates for the Graph Transformer as the key architecture for solving data scaling challenges. However, it completely overlooks the high computational and memory complexity of these models relative to traditional GNNs, thereby ignoring the critical trade-off between performance and efficiency.
3.The conclusion that Graph Transformers outperform GCNs under distribution shift is not particularly surprising, given that their powerful global attention mechanism is inherently better suited for complex scenarios. While the paper's contribution lies in connecting this to data scaling, it fails to clearly distinguish its novel perspective from the common understanding that Transformers are simply more powerful models.

**Questions:**

1.The real-world experiments in Section 3.5 simulate data scaling by using fractions of a single source graph. This tests scaling within one distribution, whereas Graph Foundation Models (GFMs) are typically pre-trained on multiple, heterogeneous domains. How do the paper's findings on data scaling generalize to this more realistic multi-domain GFM setting, where adding new domains introduces more diverse distribution shifts?
2.Your striking finding about negative data scaling is primarily demonstrated on a synthetic CSBM benchmark. Is it possible that this phenomenon is an artifact of the specific data generation process, where all source graphs are drawn from the same distribution? Have you observed similar strong negative trends on a more diverse collection of real-world pre-training datasets?

---

> ### Author Response · Authors · 2025-11-22
>
> We sincerely appreciate the reviewer’s insightful feedback and the emphasis on improving accessibility of the theoretical components.
>
> **Weakness (W1). The theoretical analysis is solid but dense; additional intuitive explanations may improve accessibility.**
>
> Response.
>
> We thank the reviewer for highlighting the need for improved accessibility. To address this, we have added a dedicated **Appendix E** with intuitive, reader-friendly explanations. The new text explains:
>
> (i) the role of the Fisher numerator (class-center separation),
>
> (ii) the meaning of Wasserstein domain misalignment,
>
> (iii) how layerwise GCN propagation amplifies or suppresses shift,
>
> (iiii) how these effects combine to produce the empirical scaling regimes.
>
> These additions in Appendix E  clarify why each assumption appears and how the theoretical bounds relate to the empirical scaling curves.
>
> **Weakness (W2).  The paper emphasizes Graph Transformers without acknowledging their higher computational cost.**
>
> Response.
>
> We appreciate this important point. We added a new **Appendix F** discussing practical efficiency considerations. Our contribution, however, concerns scaling laws under distribution shift, which are fundamentally architectural phenomena rather than efficiency considerations. We do not claim GTs should replace GNNs universally. Instead, our results show that positive data scaling requires an encoder that avoids oversmoothing and can accommodate inconsistent structure/feature cues—a condition that GTs naturally satisfy. We also note that modern implementations (FlashAttention, block-sparse attention, fused kernels) significantly reduce the overhead, and we discuss this direction as an important engineering complement.
>
> **Weakness (W3). Transformers being more powerful is expected; what is the novel insight?**
>
> Response.
>
> We thank the reviewer for this important clarification. We have added **Appendix E** emphasizing that our contribution is not to show GTs outperform GNNs, but to reveal:
>
> (i) why model scaling continues to yield gains under shift
>
> (ii) why data scaling fails under shift,
>
> (iii) which architectural properties determine whether scaling remains positive.
>
> The novelty lies in framing these behaviors as scaling laws, explaining them via Fisher and Wasserstein distance, and showing that GTs succeed because they satisfy the architectural conditions predicted by the theory—not simply because they are “powerful.”
>
> **Question (Q1). Scaling with multiple heterogeneous domains (GFM-style)**
>
> Response.
>
> We agree this is an important direction and now explicitly add it into section " Future Direction". Our study focuses on controlled single-domain scaling because: CSBM currently models controlled shifts within one domain, and  defining a principled scaling schedule across heterogeneous domains is non-trivial (e.g., sampling ratios, domain weighting, shift composition).
>
> We believe that our controlled theorical foundations—Fisher separability + Wasserstein distance + propagation—provide the conceptual tools needed for future multi-domain GFM scaling analyses.
>
> **Question (Q2). Is it possible that this phenomenon is an artifact of the specific data generation process**
>
> Response.
>
> We appreciate the reviewer’s concern. While CSBM offers a controlled environment for modulating structure–feature shifts, the negative data-scaling effect we study is not an artifact of the generative model. Multiple recent works on real graph datasets report qualitatively similar behaviors, even though they do not frame them as scaling laws.
>
> In particular, [1] shows that adding more pretraining graphs can decrease downstream performance unless the additional data are carefully selected, indicating that naïve data expansion may amplify harmful cross-domain discrepancies. More recently, [2]provides a broad empirical study across many real-world domains and finds that pretraining on additional heterogeneous datasets often fails to improve—and can even hurt—target-task accuracy. These studies present empirical evidence of the same phenomenon but do not analyze it as a scaling-law problem. Our work contributes a theoretical explanation (Fisher numerator decay + Wasserstein domain shift + propagation discrepancy) and a controlled benchmark isolating different shift sources.
>
>
> [1] Xu, Jiarong, et al. "Better with less: A data-active perspective on pre-training graph neural networks." Neurlps 2023
>
> [2] Chen Z, Mao H, Liu J, et al. Text-space graph foundation models: Comprehensive benchmarks and new insights[J]. Neurlps 2024

---

### Official Review · Reviewer_fheK · 2025-11-01

**Soundness:** 2
**Presentation:** 2
**Contribution:** 2
**Rating:** 4
**Confidence:** 3

**Summary:**

This work systematically explores scaling laws in graph self-supervised pre-training under distribution shifts, addressing the ambiguity of their applicability in graph-based pretrained models. It constructs synthetic benchmarks (via CSBM) to control structural and feature-level shifts, and conducts experiments on GNNs (e.g., GCN) and graph transformers. Key findings include: increasing model capacity consistently boosts performance, while data scaling often degrades it, though deep/wide or transformer-based architectures enable favorable data scaling. A theoretical framework based on Fisher separability and Wasserstein domain divergence explains these phenomena.

**Strengths:**

1. Provides systematic analysis of graph pre-training scaling laws under distribution shifts.
2. Combines empirical results with theoretical insights, offering both practical guidance and principled explanations.
3. Uses controllable synthetic benchmarks to disentangle structural and feature shifts.

**Weaknesses:**

1. In line 66, the authors mention "whether the absence of scaling is due to intrinsic limitations of graph models or the insufficiency of available data". In fact, there is also a lack of data. The experiments using data generated by CSBM proposed in the paper seem difficult to reflect the limitations of graph models in real-world scenarios. Although I notice the use of real datasets (USA, Europe, Brazil), these datasets are very small in scale, and the paper lacks cases with large-scale data.
2. Could the authors explore the impact of different pretraining tasks on transfer performance? Besides model architecture and parameters, the choice of graph pretraining tasks may also have a significant impact. However, the paper only seems to adopt the GraphMAE task. This could help us investigate which tasks are suitable for studying scaling laws.
3. Is the selection of structural shifts (Shift 3–5) rather simplistic? They seem to only reflect differences in data augmentation. Could CSBM be used to control the degree of structural shifts? In line 153, Shift 1 and Shift 2 appear to be indistinguishable, as both reduce inter-class separation. What is the significance of setting these two separate shifts?
4. The paper focuses primarily on node classification, which limits insights into other graph tasks (e.g., link prediction, graph classification).

**Questions:**

See weaknesses

---

> ### Author Response · Authors · 2025-11-22
>
> We thank the reviewer for the constructive feedback and the opportunity to clarify these points. Thank you for raising this important concern. In response, we substantially expanded our real-world evaluation in Appendix D.2, adding 12 additional datasets.
>
> **Weakness (W1). CSBM is synthetic and may not reflect limitations of graph models in real-world scenarios. The real datasets used (USA/Europe/Brazil) are small, and the paper lacks large-scale or diverse domain-adaptation benchmarks.**
>
> Response.
>
> Thank you for raising this important concern. In response, we substantially **expanded our real-world evaluation in Appendix D.2**, adding 12 additional datasets drawn from six recent GDA studies. Beyond the Airport networks, Appendix D.2 now includes:
> Citation: DBLPv8 → ACMv9 → Citationv2, Blog: Blog1 → Blog2, Heterogeneous ACM: ACM3 → ACM4, MAG-scale: MAG (up to 100k+ nodes). Across all dataset groups, we observe **the same scaling trends** as in CSBM. These expanded results support the external validity of our findings and directly address the reviewer’s concern regarding real-world representativeness.
>
> **Weakness (W2). The study relies mainly on GraphMAE; more diverse and recent SSL tasks should be evaluated.**
>
> Response.
>
>  We appreciate the suggestion. While the main paper highlights GraphMAE for clarity, **Appendix D.1 already includes three standard SSL families** : VGAE (generative), DGI (mutual-information), GRACE (contrastive), and GraphMAE (masked reconstruction) . During the rebuttal period, we additionally implemented four reviewer-recommended modern SSL methods **in Appendix D.1—GraphMAE2, GraphACL, HomoGCL, ReGCL**—covering spectral, homophily-aware, augmentation-free, and adversarial paradigms.
> Across all SSL objectives, we observe consistent scaling behavior, indicating that our conclusions are not tied to any specific SSL task. These additions in Appendix D.1 significantly improve completeness.
>
> **Weakness (W3). Shift design seems simplistic; Shift 1 and 2 look similar; unclear if CSBM can control shift severity.**
>
> Response.
>
> Thank you for the helpful comments. Following your suggestion, we clarified the generation process **in Appendix B**, where soft labels, prototypes, and CSBM parameters are defined explicitly. Shift 1 modifies the soft-label probability vector $p_i$, changing how each node mixes class prototypes. This models intra-class ambiguity—features become composed of more heterogeneous prototype components—while keeping both prototype means and graph structure fixed.  Shift 2 instead operates on the prototype means themselves by shrinking inter-class distances. Although both reduce separability, they represent distinct real-world feature shifts (ambiguity vs. inter-class distances shrinkage).
>
> For Shifts 3–5, we clarified that they are not augmentations but explicit CSBM-level structural manipulations:
> Shift 3: global homophily drift via altered $(p_{\mathrm{in}},p_{\mathrm{out}})$, Shift 4: local noise via edge deletions/insertions, Shift 5: combination of global + local structural shift. This adds precise control over shift severity and avoids the confounding effects seen when perturbing real graphs.
>
> **Weakness (W4). Focusing solely on node classification limits insights into link prediction and graph classification.**
>
> Response.
>
>  Thank you for the suggestion. We now explicitly acknowledge this limitation in the revised manuscript and discuss extensions to link prediction and graph classification in the **Future Directions section**. These tasks involve task-specific generators (e.g., link posteriors or graph-level prototypes) and represent promising directions for applying our framework beyond node-level transferability.

---

### Official Review · Reviewer_v8ML · 2025-11-01

**Soundness:** 2
**Presentation:** 3
**Contribution:** 3
**Rating:** 2
**Confidence:** 3

**Summary:**

This paper studies the scaling behavior of graph self-supervised pre-training under distribution shifts. The authors design controlled synthetic benchmarks using contextual stochastic block models and examine how model capacity and data scale influence transferability. They find that while increasing model capacity consistently improves performance, enlarging the pre-training dataset can hurt accuracy under distribution shifts. Through experiments on GCN, GraphSAGE, GAT, and graph transformers, the study shows that transformers exhibit more stable and positive scaling effects. The authors further propose a theoretical framework based on Fisher separability and Wasserstein divergence to explain why model capacity helps while data scaling can fail.

**Strengths:**

The paper addresses an important and timely problem which is understanding scaling laws for graph foundation models, especially under distribution shifts that are inherent to real-world graph data. The motivation is clear, the methodology is well-structured, and the experiments are described clearly. The proposed synthetic benchmark provides controlled conditions for studying the phenomenon, and empirical results show clear improvements of graph transformers over GCN-based baselines.

**Weaknesses:**

The paper a little lacks novelty to me. In detail, the experimental setup mainly repackages known components such as CSBM data, GraphMAE pretraining. The related work section is incomplete and does not sufficiently compare to concurrent studies on graph scaling or graph foundation models, graph learning under distribution shifts. The empirical evaluation includes only limited baselines, and important state-of-the-art SSL methods and graph distribution shifts methods are not fully considered. The theoretical part is more descriptive, and the connection between the derived inequalities and observed empirical trends is not convincingly supported.

**Questions:**

Can the authors clarify how the proposed theoretical framework improves over existing analyses of domain shift in GNNs? Also, since the synthetic benchmarks are highly controlled, how confident can we be that the same scaling behavior generalizes to complex real-world graph distributions beyond the small domain adaptation examples presented?

---

> ### Author Response · Authors · 2025-11-22
>
> We appreciate the reviewer’s thoughtful comments and the opportunity to clarify the motivation and positioning of our work.
>
> **Weakness (W1). The paper lacks novelty; the experimental setup mainly repackages CSBM and GraphMAE-style SSL.**
>
> Response:
> Thanks for raising this concern. Our goal is not to introduce a new model **but to establish a controlled, scalable, and theoretically interpretable environment for studying graph SSL scaling under distribution shift**—a problem for which existing real graphs are fundamentally unsuitable.
>
> Real-world graphs cannot support this study for two key reasons:
>
> **(1) No controllable generative factors.**
> Homophily level, feature prototypes, and train–test misalignment cannot be tuned without breaking semantic meaning. Large perturbations on real graphs often produce distributions that do not correspond to any valid real-world generative process, making “shift severity’’ unquantifiable.
>
> **(2) No scalability for scaling-law analysis.**
> Existing real graph datasets are limited in number, small in scale, and non-parametric. They cannot be scaled to tens of thousands of instances needed for reliable scaling curves.
>
> In contrast, **CSBM imposes no semantic constraints**, enabling parameterized manipulation of structure, features, and domain mismatch, and unlocking the theoretical components of our work (Fisher separability, spectral smoothing, GCN propagation of shift). This forms—to the best of our knowledge—the first controllable benchmark for graph SSL scaling under shift.
>
> Regarding the reviewer’s concern about “GraphMAE-style SSL’’:
> Our study is method-agnostic, and the submitted version already includes multiple, diverse SSL baselines:
>
> **(1) Included in the original submission (Appendix B.1–B.2): DGI, GRACE, VGAE**, in addition to GraphMAE.
> These are not GraphMAE-style and span contrastive, generative, and mutual-information paradigms.
>
>
> **(2) Added during rebuttal (Appendix D.1): Sp2GCL, GraphMAE2, S3GCL**
> These include augmentation-free contrastive learning, masked generative training, and homophily-aware methods.
>
> All these methods—spanning distinct related works—exhibit the same scaling behaviors under controlled shifts, reinforcing the generality of our findings and supporting that the observed phenomena are intrinsic to graph SSL rather than tied to any specific SSL method.
>
>
>
> **Weakness (W2). The related work does not sufficiently compare to graph scaling, graph foundation models, or graph distribution-shift learning.**
>
> Response:
> Thank you for pointing this out. We substantially expanded the related work in the revision:
>
> **(1) Graph scaling & graph foundation models.**
> Appendix C now provides a dedicated discussion of concurrent graph scaling papers and GFM papers. We clarify that GFMs focus on multi-task/multi-domain unification, while our work aims to build a controlled scientific environment to reveal scaling mechanisms under distribution shift—fundamentally different goals.
>
> **(2) Graph domain adaptation & distribution-shift learning.**
> We have added a detailed comparison with structural perturbation, spectral misalignment, and feature-shift methods. Most GDA works operate in supervised settings and rely on explicit alignment modules (spectral matching, feature correction, re-weighting). Our work intentionally avoids such inductive biases to expose the intrinsic scaling dynamics that arise without architectural or loss-level interventions.
>
> We hope the expanded **Appendix C** clarifies our positioning. If there are specific works the reviewer believes are especially relevant, we are happy to incorporate them.

---

> > ### Author Response · Authors · 2025-11-22
> >
> > **Weakness (W3). The empirical evaluation includes limited baselines; important SOTA SSL or shift methods are missing.**
> >
> > Response.
> >  We appreciate the reviewer’s suggestion. Although our goal is not benchmarking but understanding scaling mechanisms, we agree that including more methods improves completeness.
> >
> > **(1) On SSL baselines.**
> >  The original submission already covered four representative and widely used paradigms
> > **in Appendix B.1–B.2 : VGAE (generative), DGI (mutual information), GRACE (contrastive), and GraphMAE (masked reconstruction)**. During rebuttal, we additionally incorporated several reviewer-suggested SOTA methods, including **GraphMAE2, GraphACL, HomoGCL, and RegCL in Appendix D.1** , which span augmentation-free contrastive learning, homophily-aware design, and reconstruction-based GCL.
> > All of these methods reproduce the same scaling behaviors across structural and feature shifts, reinforcing that our conclusions are **robust, method-agnostic, and not tied to any specific SSL paradigm**.
> >
> >
> > **(2) On graph DA / distribution-shift baselines.**
> >  Existing GDA methods typically introduce alignment losses, pseudo-label refinements, or adversarial adaptation. Because these introduce additional confounders, we avoid mixing them with scaling studies and instead clarify this distinction in **Appendix C**.
> >  If the reviewer has specific GDA methods they would like to see included, we would be happy to add them.
> >
> > **Weakness (W4+Q). The theoretical part is too descriptive, and its connection to experiments is weak. How does the framework improve over existing analyses of domain shift in GNNs?**
> >
> > Response.
> >
> > Thank you for the thoughtful question. Our theoretical contribution is to introduce a unified and interpretable decomposition of target-domain Fisher separability---the key quantity controlling downstream error (Theorem 1). In the context of graph SSL pretraining, Fisher separability naturally splits into two terms:
> >  $\sqrt{F_T}\gtrsim \Delta_S(L) -\Delta_{ST}(L)$,  where both terms can be explicitly analyzed in terms of network depth, width, and the magnitude of distribution shift.
> >
> > **(1) Architecture-driven source separation.**
> > Theorem 2 shows that $\Delta_S(L)$ increases exponentially with depth and proportionally with width through the architecture factor $\theta_{\mathrm{arch}}(d)$, and improves further with larger pretraining size $N_{\mathrm{pretrain}}$. This explains why model scaling consistently enlarges source-domain class separation, and why architectures with larger $\theta_{\mathrm{arch}}(d)$---such as graph transformers with learnable aggregation---exhibit stronger gains.
> >
> > **(2) Shift-induced misalignment.**
> > Theorem 3 shows that $\Delta_{ST}(L)$ is controlled by the structural and feature Wasserstein shifts $(\varepsilon_A, \varepsilon_X)$, while depth and width reduce its effect via the contraction factor $r(d)^L$. This yields a precise trade-off: scaling helps only when the contraction induced by depth/width outweighs the looseness introduced by shift.
> >
> > **(3) Combined insight.**
> > Together, the two bounds predict the empirical phenomena in Sec. 3:
> > (i) positive model scaling when $\Delta_S(L)$ dominates;
> > (ii) diminished or saturated gains under severe shift when $\Delta_{ST}(L)$ grows large; and
> > (iii) widening architecture gaps, since transformers enjoy both a smaller propagation gap (via $r(d)$) and a larger expressive factor $\theta_{\mathrm{arch}}(d)$.
> >
> > Existing theories of GNN domain shift primarily study **supervised settings** (homophily gaps, spectral misalignment, perturbation robustness), and **do not explain how pretraining scale, architecture, and distribution shift interact**. Our framework is, to our knowledge, **the first to provide a principled mechanism for understanding why and when scaling helps or fails under distribution shift**, aligning tightly with all empirical observations in Sec.3. We have strengthened this discussion in the revised Appendix E.

---

> > > ### Author Response · Authors · 2025-11-22
> > >
> > > **Weakness (W5). Given the controlled nature of synthetic benchmarks, how confident can we be that the same scaling behavior generalizes to real-world graphs?**
> > >
> > > Response.
> > >
> > > We agree that validating the observed phenomena beyond synthetic settings is important. In the revision, we **add a set of real-world domain adaptation experiments** to complement the controlled CSBM analysis. These experiments span twelve graphs from the Airport, Citation, Blog, ACM, and MAG datasets (Sec. 3.5 & App. D.2), and capture diverse types of distribution shifts, including variations in homophily, sparsity, feature space, and cross-lingual metadata.
> > >
> > > Across the evaluated real-world transfers—such as USA→Europe, Europe→Brazil, DBLP→ACM, ACM→Citation, DE→EN, EN→FR, Blog1→Blog2, and ACM3→ACM4—we consistently observe the following:
> > >
> > > (i) Graph Transformers exhibit monotonic improvements as pretraining data increases.
> > >
> > > (ii) GCN, GAT, and GraphSAGE show less stable scaling, especially under larger distribution shifts.
> > >
> > > (iii) The performance gap between architectures widens under shift, consistent with the predictions of our theoretical decomposition.
> > >
> > > While the synthetic CSBM benchmark isolates causal factors and enables controlled analysis, these new real-world results demonstrate that **the same scaling trends hold across heterogeneous, naturally occurring graph distributions.** Full results are shown in **Appendix D.2**.

---

> ### Comment · Reviewer_v8ML · 2025-11-25
>
> Thank the authors for the responses. Some of my concerns are addressed. I appreciate the authors’ efforts, so I increased my score.

---

### Official Review · Reviewer_c78t · 2025-11-03

**Soundness:** 2
**Presentation:** 3
**Contribution:** 3
**Rating:** 4
**Confidence:** 4

**Summary:**

This work presents the systematic study of scaling laws in graph self-supervised pretraining under distribution shifts. The key finding is that increasing model capacity consistently enhances transferability, whereas the benefits of data scaling depend on model expressiveness and architecture.

**Strengths:**

1. The investigation on scaling law for graph pre-training is interesting.
2. The experimental setting is well designed.
3. The results are interesting and well justified.

**Weaknesses:**

1. The evaluation is limited to only four graph self-supervised learning (SSL) methods, all of which are not the most recent, ranging from 2016 to 2022. This makes the study less representative of the current state of graph SSL. Additionally, evaluating only four methods is insufficient to draw strong conclusions. To strengthen their claims, the authors are expected to include more recent and diverse graph SSL methods in the evaluation. Authors may consider incorporating some of the following works [1-7] for comprehensive evaluation.

[1] Bo, Deyu, et al. "Graph contrastive learning with stable and scalable spectral encoding." Advances in Neural Information Processing Systems 36 (2023): 45516-45532.

[2] Hou, Zhenyu, et al. "Graphmae2: A decoding-enhanced masked self-supervised graph learner." Proceedings of the ACM web conference 2023. 2023.

[3] Li, Wen-Zhi, et al. "Homogcl: Rethinking homophily in graph contrastive learning." Proceedings of the 29th ACM SIGKDD conference on knowledge discovery and data mining. 2023.

[4] In, Yeonjun, Kanghoon Yoon, and Chanyoung Park. "Similarity preserving adversarial graph contrastive learning." Proceedings of the 29th ACM SIGKDD Conference on Knowledge Discovery and Data Mining. 2023.

[5] Xiao, Teng, et al. "Simple and asymmetric graph contrastive learning without augmentations." Advances in neural information processing systems 36 (2023): 16129-16152.

[6] Wan, Guancheng, et al. "S3GCL: Spectral, swift, spatial graph contrastive learning." Forty-first International Conference on Machine Learning. 2024.

[7] Ji, Cheng, et al. "Regcl: Rethinking message passing in graph contrastive learning." Proceedings of the AAAI Conference on Artificial Intelligence. Vol. 38. No. 8. 2024.

**Questions:**

1. in the feature construction, can authors provide more details on how to obtain the soft labels?

2. Regarding feature perturbation, do the authors apply normalization to the node features before feeding them into the GNN during both the pre-training and linear probing stages?

---

> ### Author Response · Authors · 2025-11-22
>
> We thank the reviewer for raising this important point and fully agree that a broader set of SSL baselines strengthens the empirical claims of the paper.
>
> **Weakness. Evaluation is limited to only four older SSL methods. More recent and diverse SSL methods should be included.**
>
> Response.
> Thank you for highlighting this point. The original submission focused on four widely used paradigms—VGAE (generative), DGI (mutual-information), GRACE (contrastive), and GraphMAE (masked reconstruction)—to establish the core scaling behaviors under controlled synthetic shifts.
>
> In the revision, we add four of the reviewer-suggested modern methods—**GraphMAE2, GraphACL, HomoGCL, and RegCL**—and evaluate them across source-data sizes  $|\mathcal{D}_S| \in \[10,100,1k,5k,10k\] $ in **Appendix D.1**.  Among the seven suggested baselines, these four integrate cleanly with our pipeline and can be executed within the rebuttal timeline.
>
> Across all newly added methods, we observe the **same qualitative scaling trends** reported in the main paper:
> (i) larger models consistently yield better transfer, and
> (ii) data scaling helps only when the model’s expressiveness is sufficient relative to the shift.
>
> These additional results reinforce that our conclusions are **not tied to a narrow or outdated set of baselines**, but instead hold broadly across multiple families of modern SSL approaches.
>
> **Question (Q1). How are the soft labels obtained?**
>
> Response.
>
>  We apologize for the ambiguity caused by the term “soft label.” In our data generator, we follow standard practice in recent works on structural–feature discrepancy[1,2]. Each node is assigned a ground-truth class label $y_i \in [0,\ldots,C-1]$
> , sampled approximately uniformly (e.g., 1,000 nodes → 200 per class when $C=5$). For each class $c$, we sample a class mean $\mu_c$​, and node features are generated according to the known class-conditioned distribution.
> Thus, the “soft label” is **not produced by a classifier**, it directly comes from the **known generative process**, and is used only to control the structure–feature relationship in our synthetic benchmarks. We clarify the terminology and provide additional footnote in  the exact generation process description in **Appendix B**.
>
> [1] Baranwal, Aseem, Kimon Fountoulakis, and Aukosh Jagannath. "Graph convolution for semi-supervised classification: Improved linear separability and out-of-distribution generalization." ICML 2021
>
> [2] Mao, Haitao, et al. "Demystifying structural disparity in graph neural networks: Can one size fit all?." Neurlps 2023.
>
>
> **Question (Q2). Do you apply normalization to perturbed node features before feeding them into the GNN for pre-training and linear probing?**
>
> Response.
>
> We **do not apply additional global normalization** to the input features beyond the generative process. The same raw (possibly perturbed) features are used in both pre-training and linear probing to avoid confounding the scaling analysis. For stability, our GNN encoder uses **LayerNorm** between layers—a standard choice—kept consistent across all stages. We clarify this design choice in **Appendix B**.

---

### Author Response · Authors · 2025-12-04
**Brief summary of how we address the concerns raised by Reviewers**

**Dear Area Chair**,

We sincerely thank you for overseeing the review process and for your time and support. Below, we provide a concise summary of the revisions made during the rebuttal in direct response to reviewer feedback.

**Expanded SSL Baselines**

In response to multiple reviewers (R1, R2, R3, R4, R6), we added **four modern SSL methods—GraphMAE2, GraphACL, HomoGCL, and RegCL**—in addition to the four paradigms already in the original submission (VGAE, DGI, GRACE, GraphMAE). These additions are detailed in **Appendix D.1** and confirm that our scaling-law conclusions generalize across diverse SSL objectives.

**Real-World Generalization**

To strengthen external validity (R2, R3, R4), we significantly expanded real-world evaluation to include **12 domain-adaptation tasks** across citation, airport, blog, ACM, and MAG datasets (up to 100k+ nodes). These results (Sec. 3.5, Appendix D.2) demonstrate that the observed synthetic scaling behaviors **also hold across heterogeneous real graphs**.

**Theoretical Clarifications**

We introduced a new **Appendix E** providing intuitive explanations for our Fisher-based decomposition, domain shift propagation, and architectural factors. This directly addresses concerns from R2 and R4 about accessibility and theoretical–empirical alignment.

**Additional Architectures**

In addition to the original Graph Transformer, we included two new transformer-style models—**Nodeformer and VCR-Transformer**—during the rebuttal ( **Appendix D.3** ), confirming consistent scaling behavior across transformer families.

**Clarified Shift Design and Notation**

We revised **Appendix B** to clarify the distinct nature of synthetic shifts and how CSBM controls their severity. We also updated all real-world dataset notations (e.g., replacing “USA → EU”) for clarity (R4).

**Improved Framing and Visual Presentation**

Following R6’s feedback, we added a new **Figure 1** (overview diagram), clarified the problem formulation, and enhanced visualizations throughout (Figures 2–6, Appendix D). These changes improve readability and highlight the conceptual contributions.

**Reviewer Reactions**

While many reviewers were not able to engage directly with our responses due to the changed review process, we appreciate that one previously rejecting reviewer (R2) raised their score from 2 to 4, acknowledging that key concerns had been adequately addressed.
We believe the revised paper now provides a more complete and robust contribution, both empirically and theoretically. We hope these substantial revisions reflect our commitment to rigor and clarity, and we thank you again for your time and consideration.

Sincerely,

**The Authors**

---

### Meta-Review · Area_Chair_H1xu · 2026-01-05

**Summary:**

This paper investigates scaling law in graph self-supervised pre-training under distribution shifts. The paper reveals that merely increasing the amount of pre-training data can degrade performance, especially when the source and target graph distributions differ. The authors further demonstrate that expressive architectures—particularly deep and wide Graph Transformer models—can overcome this negative scaling effect and benefit from larger datasets. While the problem addressed is important, the reviewers have identified several notable weaknesses, including the use of limited graph self-supervised learning baselines, a narrow set of evaluation tasks, and the absence of a more realistic multi-domain graph foundation model (GFM) setting. Based on the above weaknesses, I recommend rejecting this paper.

**Reviewer Concerns:**

The rebuttal addresses some clarification requests, particularly regarding the experimental setup. However, the main concerns raised by the reviewers remain largely outstanding. In particular, the rebuttal does not expand the evaluation to a broader or more realistic set of downstream tasks, and does not sufficiently address concerns about the lack of a realistic multi-domain graph foundation model setting. As a result, the core weaknesses identified in the initial reviews persist.

**Reviewer Scores:**

The addition of several SSL baselines partially addresses the reviewer’s concerns regarding experimental completeness. However, the overall evaluation remains limited, and other major issues—such as the lack of a realistic multi-domain setting—persist.

---

### Decision · Program_Chairs · 2026-01-26

Reject